# LEAD: Large Foundation Model for EEG-Based Alzheimer's Disease Detection

## Abstract

Electroencephalogram (EEG) provides a non-invasive, highly accessible, and cost-effective solution for Alzheimer's Disease (AD) detection. However, existing methods, whether based on manual feature extraction or deep learning, face two major challenges: the lack of large-scale datasets for robust feature learning and evaluation, and poor detection performance due to inter-subject variations. To address these challenges, we curate an EEG-AD corpus containing 813 subjects, which forms the world's largest EEG-AD dataset to the best of our knowledge. Using this unique dataset, we propose **LEAD**, the first large foundation model for EEG-based AD detection. Our method encompasses an entire pipeline, from data selection and preprocessing to self-supervised contrastive pretraining, fine-tuning, and key setups such as subject-independent evaluation and majority voting for subject-level detection. We pre-train the model on 11 EEG datasets (4 AD and 7 non-AD) and unified fine-tune it on 5 AD datasets. Our self-supervised pretraining design includes sample-level and subject-level contrastive learning to extract useful general EEG features. Fine-tuning is performed on 5 channel-aligned datasets together. The backbone encoder incorporates temporal and channel embeddings to capture features across both temporal and spatial dimensions. Our method demonstrates outstanding AD detection performance, achieving up to a 9.86% increase in F1 score at the sample level and up to a 9.31% improvement at the subject level compared to state-of-the-art methods. The results of our model strongly confirm the effectiveness of subject-level contrastive pretraining and channel-aligned multi-dataset fine-tuning for addressing inter-subject variation. The source code is at https://anonymous.4open.science/r/LEAD-3B51.

[1]Anonymous Institution, Anonymous City, Anonymous Region, Anonymous Country. Correspondence to: Anonymous Author <anon.email@domain.com>.

Preliminary work. Under review by the International Conference on Machine Learning (ICML). Do not distribute.

## 1. Introduction

Alzheimer's disease (AD) is the most common neurodegenerative disorder in the elderly, affecting 10–30% of individuals over the age of 65, with an annual incidence rate of 1–3% (Breijyeh & Karaman, 2020; Masters et al., 2015). AD results from the failure to clear amyloid-$\beta$ peptide from the brain, leading to the progressive decline of cognitive functions. While there is currently no cure for AD, early intervention and treatment can slow the progression of symptoms, thereby improving patients' quality of life(Nelson & Tabet, 2015; Chu, 2012). Existing detection tools such as Magnetic Resonance Imaging (MRI) and Positron Emission Tomography (PET) neuroimaging are costly and require specialized clinical expertise, often resulting in a detection only after significant symptoms have manifested. Recently, there has been growing interest in using non-invasive techniques, such as Electroencephalogram (EEG), to identify biomarkers for AD. EEG offers real-time brain activity data and is more cost-effective than traditional methods, making it a promising tool for early detection and continuous monitoring of disease progression (Ieracitano et al., 2019a).

Currently, there are two main research directions for EEG-based Alzheimer's Disease (AD) detection. The first focuses on manually extracting feature biomarkers from the data, such as statistical features (e.g., Mean, Standard Deviation)(Tzimourta et al., 2019b;a), spectral features (e.g., Phase Shift, Phase Coherence)(Wang et al., 2017; Cassani et al., 2014), power features (e.g., Power Spectrum Density, Relative Band Power)(Fahimi et al., 2017; Schmidt et al., 2013), and complexity features (e.g., Shannon entropy, Tsallis Entropy)(Garn et al., 2015; Azami et al., 2019). Among these features, brain slowing in specific frequency bands is most commonly observed in existing research (Abásolo et al., 2005; Fahimi et al., 2017). The second direction involves using deep learning methods for automatic feature extraction. Models such as convolutional neural networks (Li et al., 2022; Cura et al., 2022), graph neural networks (Shan et al., 2022; Klepl et al., 2023), and transformers (Wang et al., 2024e) have been employed for representation learning. Some research also explores combining manual feature extraction with deep learning, such as extracting relative band powers and spectral coherence connectivity across different frequency bands and training convolutional networks on these extracted features (Miltiadous et al., 2023a).

*Figure 1.* **Pipeline of LEAD method.**

However, the detection of Alzheimer's Disease (AD) using EEG remains an open challenge, facing difficulties from both application and theoretical perspectives. From an application perspective, large, high-quality datasets are scarce. The expense and complexity of collecting EEG-AD data lead to most studies involving a limited number of subjects, often no more than 50, and typically generate only thousands of 1-second samples if segmentation is applied (Aviles et al., 2024). Such "reinventing the wheel" with self-collected small datasets causes a significant waste of resources, considering the expense of collecting EEG data from AD patients. Furthermore, the relatively small datasets used in most existing research make it difficult to demonstrate the robustness of models, limiting the generalizability of findings. From a theory perspective, the subject-independent classification in EEG-AD detection is particularly challenging due to the inter-subject variance caused by subject features interference (Wang et al., 2024c). While subjects diagnosed with AD typically should exhibit consistent patterns related to the disease, subject features such as age, gender, or other personal factors may obscure these patterns. As a result, models may overfit to these subject features rather than capturing the AD patterns (Wang et al., 2024d). This challenge is further complicated by the difficulty of interpreting EEG signals, even for experts. Unlike other domains, such as computer vision, where we can manually "remove" background features (e.g., in images), it is impossible to easily "remove" subject features from EEG data. This inherent difficulty hinders the development of methods that can effectively generalize to unseen subjects.

To address the challenges mentioned above, we propose **LEAD**, the world's first **L**arge foundational[1] model for **E**EG-based **A**lzheimer's **D**isease (AD) detection. We curate 9 EEG datasets for AD detection, both public and private, totaling 813 subjects(330 from public datasets and 483 from private sources), aiming to provide a comprehensive resource for training and evaluating detection models. **While this corpus may be relatively small compared to datasets in domains such as computer vision and natural language processing, it remains the largest EEG-based AD detection corpus to the best of our knowl-**

---

[1]In this paper, we focus on downstream tasks for EEG-Based AD detection, but our pre-trained model on different neurological disease datasets can easily extend to other brain disease detection.

**edge**. Our approach includes a full detection pipeline, including dataset selection, data preprocessing(e.g., channel and frequency alignment), self-supervised contrastive pre-training, and unified fine-tuning. We also introduce essential setups like subject-independent evaluation and majority voting for subject-level detection. The channel alignment in data preprocessing aligns all datasets into 19 standard channels, allowing us to train on different datasets. We pre-train our model on 11 datasets, which consist of 4 AD datasets and 7 additional datasets of other neurological diseases and healthy controls, including datasets for conditions like epilepsy and Parkinson's disease. This results in 2,354 subjects and 1,165,361 1-second, 128Hz samples. Our self-supervised learning design includes both sample-level and subject-level contrastive learning tasks. These tasks aim to do sample and subject discrimination, allowing the model to learn diverse EEG features that help minimize the interference of subject features in downstream tasks. We perform unified fine-tuning of the model in one run on 5 AD datasets to classify AD patients and healthy subjects, totaling 615 subjects and 223,039 1-second, 128Hz samples. We use the backbone that embeds cross-channel patches and the entire channel in parallel, capturing temporal and spatial features.

The final subject-level classification results for the 5 AD datasets—ADFTD, BrainLat, CNBPM, Cognision-ERP, and Cognision-rsEEG—are 91.34%, 89.98%, 100.00%, 84.42%, and 91.86%, respectively. We compare **LEAD** with state-of-the-art (SOTA) methods, including fully supervised, self-supervised, and EEG foundational model methods. Our results demonstrate significant improvements, with up to a 9.86% increase in F1 score at the sample level and up to a 9.31% improvement at the subject level, compared to SOTA methods. We also conduct a detailed ablation study to evaluate the impact of pre-training modules, the benefit of AD and non-AD datasets, and various training setups. Additionally, we provide supplementary studies on brain interpretability, including channel importance and frequency band analysis. **Related works** for EEG-based AD detection and self-supervised learning in EEG are in Appendix A.

We summarize our main contributions here:

- We present **LEAD**, the world's first large foundational model for EEG-based AD detection, including a comprehensive method pipeline.
- We construct the world's largest EEG-based AD detection corpus, consisting of 9 datasets with 813 subjects.
- Our strong performance validates the effectiveness of subject-level contrastive pre-training and unified fine-tuning for EEG-based AD detection.
- We release our code and model checkpoints to break the isolation in the EEG-based AD detection domain and facilitate future research.

## 2. Method

### 2.1. Problem Formulation

**Sample-Level Classification.** Consider an input EEG sample $x \in \mathbb{R}^{T \times C}$ where $T$ denotes the number of timestamps and $C$ represents the number of channels. Our objective is to learn an encoder that generates a representation $h$ which can be used to predict the corresponding label $y \in \mathbb{R}$ for the input sample. Specifically, the label $y$ corresponds to either Alzheimer's Disease or Healthy controls.

**Subject-Level Classification.** In addition to the corresponding label $y \in \mathbb{R}$ each input EEG sample also has a subject ID $s \in \mathbb{R}$ that indicates which subject the sample belongs to. The ultimate goal of EEG-based AD detection is to determine whether a subject has Alzheimer's Disease. For subject-level classification, we use a majority voting scheme, where the subject is assigned the label corresponding to the majority label of all samples from that subject.

### 2.2. Datasets Selection

**AD Datasets.** We review EEG-based AD detection papers published between 2018 and 2024 to identify potentially available datasets. We find 6 publicly available datasets containing AD subjects: **AD-Auditory**(Lahijanian et al., 2024), **ADFSU**(Vicchietti et al., 2023), **ADFTD**(Miltiadous et al., 2023b;a), **ADSZ**(Alves et al., 2022; Pineda et al., 2020), **APAVA**(Escudero et al., 2006; Smith et al., 2017), and **BrainLat**(Prado et al., 2023). Additionally, we use 3 private datasets: **Cognision-ERP**(Cecchi et al., 2015), **Cognision-rsEEG**, and **CNBPM**(Ieracitano et al., 2019b; Amezquita-Sanchez et al., 2019), bringing the total number of AD datasets to 9, and the total number of subjects to 813. We perform preliminary experiments on each dataset individually to assess their quality. For smaller datasets or those showing large performance variability across subjects, we use them for pre-training to alleviate potential data quality issues such as mislabeled subjects, interference from artifacts, collection devices, and collection methods. Five high-quality AD datasets, **ADFTD**, **CNBPM**, **Cognision-rsEEG**, **Cognision-ERP**, and **BrainLat**, are used for downstream tasks to evaluate the model performance.

**Non-AD Datasets.** To enhance the learning of general EEG and AD-specific features, we use datasets of healthy subjects and other neurological diseases for self-supervised pretraining. We aim to increase the diversity of brain conditions, including healthy and diseased states, and increase the number of subjects used for training to reduce the interference of subject-specific patterns. Note that all the non-AD datasets have one commonality: the label is assigned to the subject, which adapts to the subject-level feature extraction. Datasets such as sleep stage detection and mental state classification are unsuitable here. We select publicly available datasets from sources like OpenNEURO[2], Temple University Hospital[3], and Brainclinics[4]. We choose datasets collected in a resting-state condition or involving resting-state tasks with either eyes open or closed to ensure consistency with most downstream AD datasets. In total, we select 7 proper large datasets, each with hundreds or even thousands of subjects. They are **Depression** (Cavanagh et al., 2019; Cavanagh, 2021), **PEARL-Neuro** (Dzianok & Kublik, 2024), **REEG-BACA** (Getzmann et al., 2024), **REEG-PD** (Singh et al., 2023), **REEG-SRM** (Hatlestad-Hall et al., 2022), **TDBrain** (Van Dijk et al., 2022), and **TUEP** (Veloso et al., 2017).

### 2.3. Data Preprocessing

Two key challenges in training a large foundation model for time-series-like data are varying channel/variate numbers and heterogeneous sampling frequencies (Liu et al., 2024; Woo et al., 2024; Yang et al., 2024). However, we can easily align channels based on their names in EEG and align sampling frequency by resampling. More details and reasons for preprocessing steps are provided in Appendix D.

**Artifacts Removal.** Some datasets have already undergone preprocessing steps during data collection, such as artifact removal and filtering. We perform a secondary preprocessing to align all datasets uniformly for training. All the fine-tuning datasets are guaranteed to be artifacts-free.

**Channel Alignment.** We align all datasets to a standard set of 19 channels, which include Fp1, Fp2, F7, F3, Fz, F4, F8, T3/T7, C3, Cz, C4, T4/T8, T5/P7, P3, Pz, P4, T6/P8, O1, and O2, based on the international 10-20 system[5]. For datasets with fewer than 19 channels, we interpolate the missing channels using the MNE EEG processing package[6]. For datasets with more than 19 channels, we select the 19 channels based on the channel name and discard the others. In cases where datasets use different channel montages, such as the Biosemi headcaps with 32, 64, 128 channels[7], we select the 19 closest channels by calculating the Euclidean distance between their 3D coordinates. The channel alignment allows us to pre-train the models on different datasets with any backbone encoder and perform unified fine-tuning on all AD datasets in one run.

**Frequency Alignment.** In addition to channel alignment, we resample all datasets to a uniform sampling frequency of 128Hz, which is commonly used and preserves the key frequency bands ($\delta, \theta, \alpha, \beta, \gamma$), while also reducing noise.

---

[2]https://openneuro.org/
[3]https://isip.piconepress.com/projects/tuh_eeg/
[4]https://www.brainclinics.com/resources
[5]https://en.wikipedia.org/wiki/10-20_system_(EEG)
[6]https://mne.tools/stable/index.html
[7]https://www.biosemi.com/headcap.htm

**Sample Segmentation.** For deep learning training, we segment the EEG trials within each subject into 1-second samples, which results in 128 timestamps per sample, as the sampling frequency is aligned to 128Hz.

**Frequency Filtering.** We then apply frequency filtering to each sample, ranging from 0.5Hz to 45Hz, to remove frequency bands that do not correspond to brain activities.

**Standard Normalization.** After frequency filtering, we perform standard normalization on each sample, applied individually to each channel, to ensure that the data is centered and scaled consistently across all samples and channels.

### 2.4. Self-Supervised Pretraining

The region b) in figure 2 shows the flowchart of self-supervised contrastive pre-training.

**Representation Learning.** For an input EEG sample $x_i$, where $i$ denotes the index of the sample $x_i$, we apply data augmentation methods $a$ and $b$ to generate two augmented views, $x_i^a$ and $x_i^b$. Given a backbone encoder $f(\cdot)$ and a projection head $g(\cdot)$, we compute their representations $h_i^a = f(x_i^a)$ and $h_i^b = f(x_i^b)$ after the encoder $f(\cdot)$, and further obtain denser representations $z_i^a = g(h_i^a)$ and $z_i^b = g(h_i^b)$ through the projection head $g(\cdot)$. The projection head is designed to benefit contrastive learning, as described in (Chen et al., 2020), and it will be discarded during downstream tasks, with only the encoder being used for the downstream task.

**Sample-Level Contrasting.** Sample-level contrasting is the most widely used framework in contrastive learning, as seen in SimCLR (Chen et al., 2020) and MOCO (He et al., 2020). The goal is to perform sample/instance discrimination and learn a representation that can distinguish one sample from others (Wu et al., 2018). A pre-trained model using this approach can capture general patterns in EEG data, benefiting downstream tasks by improving performance and reducing the need for labeled data. In this work, we adopt the SimCLR architecture, which treats different augmented views of the same sample as positive pairs and views from different samples as negative pairs. For an input sample $x_i \in \mathcal{B}$ in a batch, our sample-level InfoNCE contrastive loss is defined as follows:

$$\mathcal{L}_{Sam} = \mathbb{E}_{x_i} \left[ -\log \frac{\exp(\text{sim}(z_i^a, z_i^b)/\tau)}{\sum_j \left( \exp(\text{sim}(z_i^a, z_j^b)/\tau) \right)} \right] \quad (1)$$

where $j$ denotes the index of other samples in the batch $\mathcal{B}$, and $\text{sim}(u, v) = \frac{u^T v}{\|u\|\|v\|}$ denotes the cosine similarity between vectors $u$ and $v$. The parameter $\tau$ is a temperature parameter that adjusts the similarity scale.

**Subject-Level Contrasting.** In EEG-based Alzheimer's disease (AD) detection, each subject is typically associated with a stable medical state. Specifically, once a subject has AD or preclinical signs of AD, all EEG samples from that subject should exhibit features related to AD, meaning they share the same label during deep learning training. This prior knowledge allows us to perform subject-level contrasting, a concept first defined in (Wang et al., 2024b) and successfully applied in EEG and ECG-based disease detection (Kiyasseh et al., 2021; Wang et al., 2024b; Abbaspourazad et al., 2024). In subject-level contrasting, we treat samples from the same subject as positive pairs and samples from different subjects as negative pairs. With an increasing number of subjects used in pre-training, we aim for the model to learn diverse feature types and reduce interference from unrelated subject-specific features during downstream classification. Appendix F.3 and H.2 provide more details on the effectiveness and analysis of subject-level contrasting. For an input sample $x_i \in \mathcal{B}$ in a batch, our subject-level InfoNCE contrastive loss is defined as follows:

$$\mathcal{L}_{Sub} = \mathbb{E}_{x_i} \left[ \mathbb{E}_{x_k} \left[ -\log \frac{\exp(\text{sim}(z_i^a, z_k^b)/\tau)}{\sum_j \left( \exp(\text{sim}(z_i^a, z_j^b)/\tau) \right)} \right] \right] \quad (2)$$

where $x_k$ denotes samples from the same subject as $x_i$ in the batch, with the same subject ID $s_k = s_i$. The function $\text{sim}(u, v) = \frac{u^T v}{\|u\|\|v\|}$ represents the cosine similarity, and $\tau$ is a temperature parameter that adjusts the scale. Note that not all neurological diseases can utilize subject-level contrasting. For instance, seizures are a condition where the EEG patterns during a seizure phase differ significantly from those in the regular phase for the same subject.

**Overall Loss Function.** The overall loss function is the weighted sum of the sample-level and subject-level contrastive losses is defined as follows:

$$\mathcal{L} = \lambda_1 \mathcal{L}_{Sam} + \lambda_2 \mathcal{L}_{Sub} \quad (3)$$

where $\lambda_1 + \lambda_2 = 1$ are hyper-coefficients that control the relative importance and adjust the scales of each level's loss.

**Indices Shuffling.** In real-world scenarios, the likelihood of samples with the same subject ID appearing in the same training batch decreases as the number of subjects increases. This can hinder subject-level contrastive learning. To address this issue, we develop an indices shuffling algorithm that shuffles the order of samples in each epoch. The goal is to ensure that samples with the same subject ID are present in the batch while introducing randomness in the sample order every epoch. More algorithm description and Pseudo code details are presented in Appendix B.

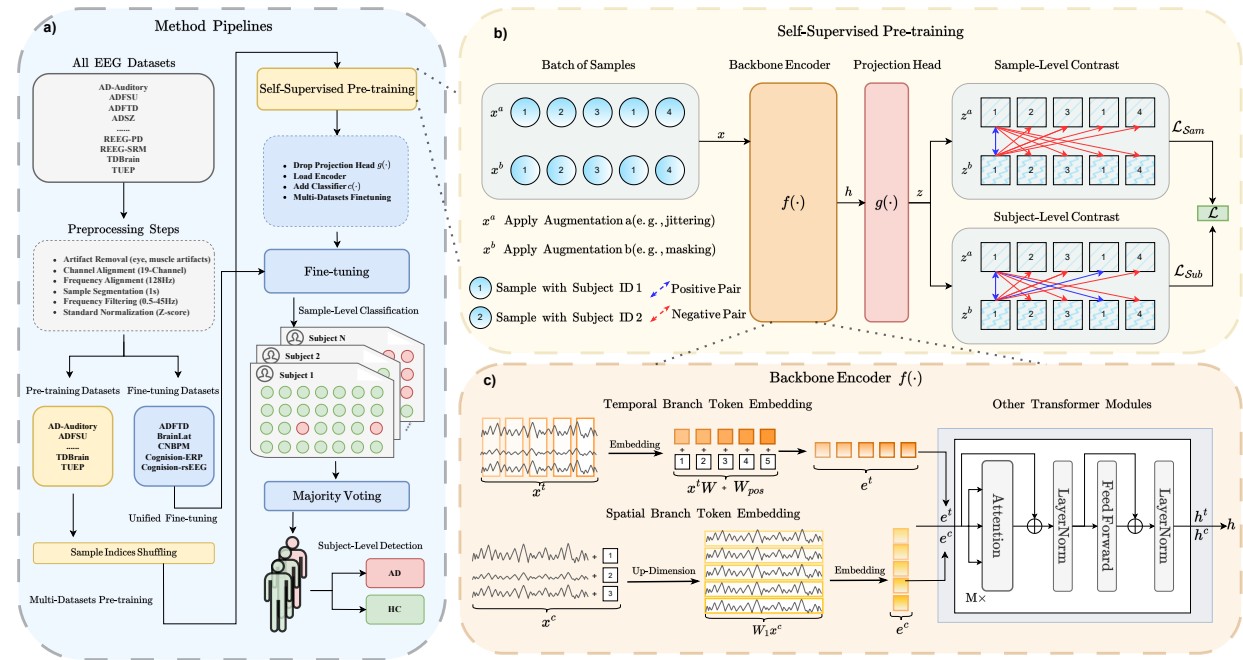

*Figure 2.* **Details of LEAD method. a)** The pipeline for our method includes data preprocessing, sample indices shuffling, self-supervised pre-training, multi-dataset fine-tuning, sample-level classification, and subject-level AD detection. **b)** The flowchart of the self-supervised pre-training. A batch of samples is applied with two augmentations $a$ and $b$ to generate two augmented views, $\boldsymbol{x}_i^a$ and $\boldsymbol{x}_i^b$. The number in each sample is the subject ID. The representation $\boldsymbol{z}_i^a$ and $\boldsymbol{z}_i^b$ after encoder $f(\cdot)$ and projection head $g(\cdot)$ are used for contrastive learning. Two augmented views from the same sample are positive pairs for sample-level contrast. For subject-level contrast, samples with the same subject IDs are positive pairs. **c)** The backbone encoder $f(\cdot)$ includes two branches. The temporal branch takes cross-channel patches to embed as tokens. The spatial branch takes the whole series of channels to embed as tokens. The two branches are computed in parallel.

## 2.5. Backbone Encoder Architecture

We use a simplified version of ADformer (Wang et al., 2024e) as the backbone encoder $f(\cdot)$, adopting single-granularity learning only. This architecture is designed for EEG-based AD detection and efficiently captures temporal features along the time dimension and spatial features among channels, as both are critical for EEG feature representation learning. For simplicity, we omit the subscript $i$ for the input sample $\boldsymbol{x}$ in this subsection, as it is not necessary for the illustration. The temporal and spatial branches are computed in parallel before the projection head or classifier. Both branches use the standard encoder-only transformer, including self-attention, layer normalization, and feed-forward networks. The region c) in figure 2 illustrates the architecture of the backbone encoder.

**Temporal Branch.** Given an input EEG sample $\boldsymbol{x} \in \mathbb{R}^{T \times C}$ and patch length $L$, where $T$ and $C$ denote the number of timestamps and channels, respectively. We first segment the input sample into $N$ cross-channel non-overlapping patches to obtain $\boldsymbol{x}^{\mathrm{t}} \in \mathbb{R}^{N \times (L \cdot C)}$. Zero padding is applied to ensure that the number of timestamps $T$ is divisible by $L$, resulting in $N = \lceil \frac{T}{L} \rceil$. The patches $\boldsymbol{x}^{\mathrm{t}}$ are then mapped into $D$-dimensional patch embeddings using a linear projection $\boldsymbol{W}$, and a fixed positional embedding $\boldsymbol{W}_{\mathrm{pos}}$ (Vaswani et al.,

2017) is added to produce the final patch embeddings: $\boldsymbol{e}^{\mathrm{t}} = \boldsymbol{x}^{\mathrm{t}}\boldsymbol{W} + \boldsymbol{W}_{\mathrm{pos}}$, where $\boldsymbol{e}^{\mathrm{t}} \in \mathbb{R}^{N \times D}$, $\boldsymbol{W} \in \mathbb{R}^{(L \cdot C) \times D}$, and $\boldsymbol{W}_{\mathrm{pos}} \in \mathbb{R}^{N \times D}$. The final patch embeddings $\boldsymbol{e}^{\mathrm{t}}$ are used as input tokens for the standard encoder-only transformer. After $M$ encoding layers, we obtain the temporal branch's final representations $\boldsymbol{h}^{\mathrm{t}}$.

**Spatial Branch.** Given an input EEG sample $\boldsymbol{x} \in \mathbb{R}^{T \times C}$, we first transpose the sample and add a fixed channel-wise positional embedding $\boldsymbol{W}_{\mathrm{pos}}$ to obtain $\boldsymbol{x}^{\mathrm{c}} = \mathrm{Transpose}(\boldsymbol{x}) + \boldsymbol{W}_{\mathrm{pos}}$, where $\boldsymbol{x}^{\mathrm{c}}, \boldsymbol{W}_{\mathrm{pos}} \in \mathbb{R}^{C \times T}$. Unlike the temporal branch, where positional embeddings are added after embedding, we add channel-wise positional embeddings on the raw input EEG data since the subsequent up-dimension process destroys the information of raw channel order. For a target channel number $F$ and embedding dimension $D$, we first perform an up-dimensional transformation using a 1-D convolution $\boldsymbol{W}_1$ to increase the channel number. Then, we map the entire series of each channel into a latent embedding using a linear projection $\boldsymbol{W}_2$ to get the final channel embeddings: $\boldsymbol{e}^{\mathrm{c}} = (\boldsymbol{W}_1 \boldsymbol{x}^{\mathrm{c}}) \boldsymbol{W}_2$, where $\boldsymbol{e}^{\mathrm{c}} \in \mathbb{R}^{F \times D}$, $\boldsymbol{W}_1 \in \mathbb{R}^{F \times C}$, and $\boldsymbol{W}_2 \in \mathbb{R}^{T \times D}$. The final channel embeddings $\boldsymbol{e}^{\mathrm{c}}$ are used as input tokens for the standard encoder-only transformer. After $M$ encoding layers, we obtain the spatial branch's final representation $\boldsymbol{h}^{\mathrm{c}}$.

**Projection Head and Classifier.** For an input EEG sample $x$, we obtain the temporal branch's representation $h^t$ and the spatial branch's representation $h^c$. We concatenate the **last token** from both representations to form the final representation $h = [h^t[-1] \,||\, h^c[-1]]$ of the backbone encoder $f(\cdot)$, where $[\cdot \,||\, \cdot]$ denotes concatenation and $h \in \mathbb{R}^{2D}$. For contrastive pre-training, $h$ is further projected into a denser representation $z \in \mathbb{R}^D$ using a projection head $g(\cdot)$, where $g(\cdot)$ consists of a two-layer fully connected network. For downstream classification tasks, $h$ is used directly to classify the output label $y$ via a linear classifier $c(\cdot)$.

### 2.6. Important Setups

**Subject-Independent.** Two main setups are commonly used for evaluation in the EEG-based AD detection domain: subject-dependent (Nour et al., 2024; kumar Ravikanti & Saravanan, 2023) and subject-independent (Watanabe et al., 2024; Chen et al., 2024). In the subject-dependent setup, all samples are mixed together and split into training, validation, and test sets, allowing samples from the same subject to appear in all three sets. In contrast, the subject-independent setup splits the training, validation, and test sets based on subjects, ensuring that samples from the same subject are exclusively assigned to one set (Wang et al., 2024c). Unlike many existing works that use the subject-dependent setup, we use the subject-independent setup. The subject-dependent setup is unsuitable for real-world scenarios and leads to significant data leakage (Wang et al., 2024d).

**Unified Fine-tuning.** The channel alignment in our data preprocessing step enables us to pre-train the model on various datasets and then fine-tune it on all downstream datasets simultaneously. We refer to this as "unified fine-tuning," where the model is fine-tuned across all downstream AD datasets in one run. The best model is then selected based on the weighted performance across the downstream datasets, ensuring that the model performs optimally on all tasks.

**Majority Voting.** For subject-level EEG-based AD detection, we apply a majority voting scheme to determine the final classification label for each subject. Specifically, for all the samples from one subject (with the same subject ID $s$), we find the majority label of these samples and assign this label to this subject. For example, if a subject has 100 samples and more than 50 are classified as AD, the subject will be labeled "AD." The voting mechanism alleviates the interference of outlier samples in a subject.

## 3. Experiments

**Datasets.** We pre-train on 11 datasets: **AD-Auditory** (Lahijanian et al., 2024), **ADFSU** (Vicchietti et al., 2023), **ADSZ** (Pineda et al., 2020), **APAVA** (Escudero et al., 2006), **Depression** (Cavanagh et al., 2019), **PEARL-**

**Neuro** (Dzianok & Kublik, 2024), **REEG-BACA** (Getzmann et al., 2024), **REEG-PD** (Singh et al., 2023), **REEG-SRM** (Hatlestad-Hall et al., 2022), **TDBrain** (Van Dijk et al., 2022), and **TUEP** (Veloso et al., 2017), and fine-tuning on 5 downstream datasets: **ADFTD** (Miltiadous et al., 2023b), **BrainLat** (Prado et al., 2023), **CNBPM** (Amezquita-Sanchez et al., 2019), **Cognision-ERP** (Cecchi et al., 2015), and **Cognision-rsEEG**. The pre-training datasets include 7 non-AD neurological diseases or healthy subjects and 4 AD datasets, totaling *2,354 subjects and 1,165,361 1-second, 128Hz samples*. All downstream datasets are binary classifications between AD patients and healthy subjects, totaling *615 subjects and 223,039 1-second, 128Hz samples*. The nine AD datasets used for pretraining or fine-tuning consist of *813 subjects* in total. The rationale behind selecting these datasets for pre-training and fine-tuning is discussed in 2.2. The unified processing pipeline for each dataset is detailed in 2.3, with a more detailed description available in Appendix D. The statistics for the processed datasets are summarized in Table 1.

**Baselines.** We compare our method with 10 baselines, including 5 supervised, 3 self-supervised learning, and 2 large EEG foundational models. These selected baselines are state-of-the-art methods or have shown strong performance in EEG or time series classification tasks. The 5 supervised learning methods include **TCN** (Bai et al., 2018), vanilla **Transformer** (Vaswani et al., 2017), **Conformer** (Song et al., 2022), **TimesNet** (Wu et al., 2023), and **Medformer** (Wang et al., 2024c). The 3 self-supervised learning methods are **TS2Vec** (Yue et al., 2022), **BIOT** (Yang et al., 2024), and **EEG2Rep** (Mohammadi Foumani et al., 2024). The 2 large EEG foundational models are **LaBraM** (Jiang et al., 2024) and **EEGPT** (Wang et al., 2024a).

**Implementation.** All baseline methods and our method's variants, except for LaBraM and EEGPT, are trained under the same code framework. The training epoch for self-supervised pretraining is fixed at 50 epochs, with no early stopping mechanism. The training epoch is set to 100 for fully supervised learning or fine-tuning, with early stopping after 15 epochs of patience based on the best F1 score. The batch sizes for pretraining, fully supervised learning, and fine-tuning are set to 512, 128, and 128, respectively. The optimizer is AdamW. The initial learning rates for pretraining, fully supervised learning, and fine-tuning are set to 0.0002, 0.0001, and 0.0001, respectively, with the CosineAnnealingLR learning scheduler. Gradient norm clipping is set to 4.0, and Stochastic Weight Averaging (SWA) (Izmailov et al., 2018) is enabled to benefit inter-subject representation learning. For LaBraM and EEGPT, we use their public code and load their pre-trained model for fine-tuning. We employ four evaluation metrics: sample-level accuracy and F1 score (macro-averaged), and subject-level accuracy and F1 score (macro-averaged) after majority voting, as described in 2.6.

*Table 1.* **Processed Dataset Statistics.** For datasets where the channels are not 19 standard channels in the international 10-20 system, we process them into two versions: one with channel alignment to the 19 channels and one without channel alignment. We present the #-AD and #-HC subjects in the supervised or unified fine-tuning task, *e.g.*, 35 AD + 29 HC = 65 subjects.

| Datasets | Confidentiality | Type(subtype) | #-Subjects | #-Rate | #-Channels | #-Duration | #-Samples | Tasks |
|---|---|---|---|---|---|---|---|---|
| **AD-Auditory** | Public | AD(ERP) | 35 | 128Hz | 19 | 1 second | 37,425 | Self-supervised pre-training |
| **ADFSU** | Public | AD(Resting) | 92 | 128Hz | 19 | 1 second | 2,760 | Self-supervised pre-training |
| **ADSZ** | Public | AD(Resting) | 48 | 128Hz | 19 | 1 second | 768 | Self-supervised pre-training |
| **APAVA** | Public | AD(Resting) | 23 | 128Hz | 16 or 19 | 1 second | 5,967 | Self-supervised pre-training |
| **ADFTD** | Public | AD(Resting) | 36+29=65 | 128Hz | 19 | 1 second | 53,215 | Supervised or Unified Fine-tuning |
| **BrainLat** | Public | AD(Resting) | 35+32=67 | 128Hz | 19 | 1 second | 29,788 | Supervised or Unified Fine-tuning |
| **CNBPM** | Private | AD(Resting) | 63+63=126 | 128Hz | 19 | 1 second | 46,336 | Supervised or Unified Fine-tuning |
| **Cognision-rsEEG** | Private | AD(Resting) | 97+83=180 | 128Hz | 7 or 19 | 1 second | 32,400 | Supervised or Unified Fine-tuning |
| **Cognision-ERP** | Private | AD(ERP) | 90+87=177 | 128Hz | 7 or 19 | 1 second | 61,300 | Supervised or Unified Fine-tuning |
| **Depression** | Public | Non-AD(Resting) | 122 | 128Hz | 66 or 19 | 1 second | 24,014 | Self-supervised pre-training |
| **PEARL-Neuro** | Public | Non-AD(Resting) | 79 | 128Hz | 127 or 19 | 1 second | 51,670 | Self-supervised pre-training |
| **REEG-BACA** | Public | Non-AD(Resting) | 608 | 128Hz | 65 or 19 | 1 second | 611,269 | Self-supervised pre-training |
| **REEG-PD** | Public | Non-AD(Resting) | 149 | 128Hz | 60 or 19 | 1 second | 23,839 | Self-supervised pre-training |
| **REEG-SRM** | Public | Non-AD(Resting) | 109 | 128Hz | 64 or 19 | 1 second | 32,760 | Self-supervised pre-training |
| **TDBrain** | Public | Non-AD(Resting) | 911 | 128Hz | 33 or 19 | 1 second | 231,689 | Self-supervised pre-training |
| **TUEP** | Public | Non-AD(Resting) | 179 | 128Hz | 19 | 1 second | 143,200 | Self-supervised pre-training |

In the self-supervised pre-training stage, all subjects in the datasets are used for training. The $\lambda_1$ and $\lambda_2$ are both set to 0.5. In the supervised learning or fine-tuning classification stage, the training, validation, and test sets are split based on the subject-independent setup with a ratio of 6:2:2 for each dataset, where each subject appears exclusively in one of these three sets. There is no dataset overlapping between the pre-training and fine-tuning datasets. The training process is conducted with 5 random seeds (41-45) on fixed training, validation, and test sets to compute the mean and standard deviation of the models. All experiments are run on an RTX 4090 GPU and a server with 4 RTX A5000 GPUs, using Python 3.8 and PyTorch 2.0.0 + cu118. Appendix E provides more details about each method's implementations.

### 3.1. Comparison with Baselines

**Setup.** Our method has three variants based on training setups: **LEAD-Vanilla(3.21M)**, **LEAD-Sup(3.21M)**, and **LEAD-Base(3.41M)**. The LEAD-Vanilla model is trained fully supervised on a single dataset without channel alignment, such as the 7-channel version of the Cognision-ERP dataset. LEAD-Sup and LEAD-Base use datasets with alignment to 19 channels. LEAD-Sup is the model trained unified supervised on 5 AD datasets together without pre-training. For LEAD-Base, we first perform self-supervised pre-training on 11 pre-training datasets. The trained model is then used for unified fine-tuning on 5 downstream AD datasets. Note that for both LEAD-Sup and LEAD-Base, the 5 downstream AD datasets are unified trained and evaluated in one run, which is different from the usual approach where supervised training or fine-tuning occurs on a single dataset. The five supervised learning baselines, including TCN, Transformer, Conformer, TimesNet, and Medformer, use the same setup as LEAD-Vanilla. The three self-supervised learning baselines, including TS2Vec, BIOT,

and EEG2Rep, follow LEAD-Base's setup. For the two large EEG foundational models, LaBraM and EEGPT, we load their pre-trained models and use the same fine-tuning setup as our LEAD-Base. Appendix E provides more details about the implementation setups.

**Results.** The results are presented in Table 2. Our method significantly improves accuracy and F1 score compared with all baselines for both sample-level and subject-level classification. Specifically, our method outperforms the best baseline methods by 6.9%, 5.72%, 3.85%, 7.81%, and 11.16% in F1 score at the subject-level on the ADFTD, BrainLat, CNBPM, Cognision-ERP, and Cognision-rsEEG datasets, respectively. The comparison between our method and the supervised learning baselines highlights the effectiveness of channel alignment; although some information might be lost during alignment, the ability to allow unified training still demonstrates substantial performance improvements compared to supervised learning methods on raw-channel datasets. The comparison with self-supervised learning baselines underscores the effectiveness of our contrastive learning approach. The sample-level and subject-level contrasting show a strong learning ability for inter-subject classification. The two large EEG models perform poorly on the ADFTD and BrainLat datasets, achieving almost random results. The comparison between ours and their methods emphasizes the importance of selecting proper pre-training datasets. Our selection of healthy and neurological disease datasets for pre-training contributes significantly to the downstream classification between AD and healthy controls.

Among the three variants of our methods, the LEAD-Base achieves the best performance in most cases, except for the ADFTD dataset, where LEAD-Sup performs better. The comparison between LEAD-Vanilla and LEAD-Sup shows that leveraging more AD datasets for training benefits performance, even in a fully supervised learning manner. The

*Table 2.* **Comparison with Baselines.** This table presents the sample and subject-level classification results of 10 baselines and 4 variations of our method. The parameter number of each method is written in the brackets, followed by the method name.

| Datasets | ADFTD (53,215 Samples) (65 Subjects) | | BrainLat (29,788 Samples) (67 Subjects) | | CNBPM (46,336 Samples) (126 Subjects) | | Cognision-ERP (61,300 Samples) (177 Subjects) | | Cognision-rsEEG (32,400 Samples) (180 Subjects) | |
|---|---|---|---|---|---|---|---|---|---|---|
| **Sample-Level Classification** | | | | | | | | | | |
| Metrics / Methods | Accuracy | F1 Score | Accuracy | F1 Score | Accuracy | F1 Score | Accuracy | F1 Score | Accuracy | F1 Score |
| **TCN(1.02M)** | 75.01±0.95 | 74.90±0.90 | 59.23±0.97 | 59.21±0.98 | 92.35±0.33 | 90.28±0.40 | 61.87±0.45 | 61.43±0.67 | 64.27±0.82 | 63.78±0.89 |
| **Transformer(0.83M)** | 67.89±2.14 | 67.29±2.53 | 59.21±1.05 | 58.60±0.83 | 93.37±0.25 | 91.74±0.33 | 60.56±0.25 | 60.42±0.23 | 59.58±0.32 | 59.52±0.27 |
| **Conformer(1.14M)** | 75.04±1.56 | 74.80±1.66 | 62.45±1.09 | 60.76±2.19 | 86.50±2.36 | 84.80±2.34 | 63.38±0.21 | 63.33±0.19 | 65.36±0.34 | 64.78±0.49 |
| **TimesNet(2.35M)** | 75.06±1.59 | 74.42±2.07 | 59.63±2.16 | 59.09±2.49 | 92.28±0.52 | 90.13±0.70 | 62.04±0.31 | 61.93±0.38 | 61.48±0.84 | 60.60±0.79 |
| **Medformer(2.42M)** | 73.88±1.23 | 73.77±1.19 | 60.15±0.79 | 59.86±0.86 | 93.73±0.29 | 92.22±0.37 | 61.23±0.57 | 60.58±0.74 | 62.89±0.75 | 62.50±0.73 |
| **TS2Vec(2.58M)** | 71.81±0.84 | 71.73±0.83 | 67.99±1.10 | 67.94±1.08 | 91.79±0.67 | 89.38±0.94 | 62.19±0.67 | 62.03±0.75 | 67.08±0.44 | 66.76±0.46 |
| **BIOT(4.16M)** | 78.63±0.80 | 77.43±0.89 | 61.51±1.50 | 61.36±1.27 | 88.31±0.46 | 83.56±0.88 | 63.41±0.26 | 63.14±0.32 | 66.14±1.23 | 65.41±1.64 |
| **EEG2Rep(5.33M)** | 70.62±1.31 | 70.60±1.32 | 68.02±3.86 | 67.60±3.87 | 91.41±1.31 | 88.96±1.97 | 64.04±0.74 | 63.92±0.68 | 70.12±1.52 | 69.85±1.66 |
| **LaBraM(9.62M)** | 55.07±0.00 | 71.03±0.00 | 48.24±0.00 | 65.08±0.00 | 78.44±1.92 | 83.46±1.63 | 70.90±1.52 | 72.92±1.22 | 57.16±1.47 | 61.21±3.00 |
| **EEGPT(25.5M)** | 54.09±0.00 | 70.21±0.00 | 49.09±0.00 | 65.85±0.00 | 68.44±3.44 | 74.53±2.14 | 59.58±3.17 | 64.60±1.69 | 57.66±0.86 | 62.23±2.42 |
| **LEAD-Vanilla(3.21M)** | 73.81±1.02 | 73.75±1.00 | 62.15±1.28 | 62.07±1.28 | 94.94±0.20 | 93.65±0.26 | 62.64±0.86 | 62.56±0.86 | 61.43±1.41 | 60.98±1.48 |
| **LEAD-Sup(3.21M)** | **80.84±0.84** | **80.68±0.79** | 70.36±0.62 | 70.31±0.65 | 94.24±0.46 | 92.65±0.61 | 65.96±0.92 | 65.93±0.92 | 71.72±0.69 | 71.41±0.71 |
| **LEAD-Base(3.41M)** | 76.64±0.87 | 76.64±0.86 | **77.89±1.28** | **77.80±1.34** | **96.51±0.33** | **95.53±0.42** | **69.58±0.90** | **69.53±0.91** | **76.21±0.39** | **76.01±0.39** |
| **Subject-Level Classification** | | | | | | | | | | |
| Metrics / Methods | Accuracy | F1 Score | Accuracy | F1 Score | Accuracy | F1 Score | Accuracy | F1 Score | Accuracy | F1 Score |
| **TCN(1.02M)** | 81.43±3.50 | 81.36±3.55 | 75.71±7.28 | 75.57±7.24 | 96.15±0.00 | 96.15±0.00 | 71.11±2.22 | 70.70±2.29 | 71.35±3.67 | 70.42±4.26 |
| **Transformer(0.83M)** | 74.29±3.50 | 74.00±3.65 | 74.29±5.71 | 72.82±6.12 | 90.77±1.88 | 90.73±1.87 | 67.22±2.72 | 66.98±2.84 | 71.35±3.67 | 70.95±3.89 |
| **Conformer(1.14M)** | 77.14±5.35 | 76.86±5.59 | 68.57±11.61 | 63.40±15.55 | 92.31±0.00 | 92.26±0.00 | 69.44±1.76 | 68.49±1.82 | 71.35±2.16 | 70.48±2.49 |
| **TimesNet(2.35M)** | 81.43±3.50 | 81.05±3.18 | 77.14±5.35 | 76.23±5.90 | 86.15±3.08 | 86.12±3.09 | 70.56±2.83 | 70.40±2.83 | 69.19±2.76 | 67.30±3.18 |
| **Medformer(2.42M)** | 78.57±0.00 | 78.46±0.00 | 74.29±5.71 | 73.51±5.37 | 92.31±0.00 | 92.26±0.00 | 67.22±1.11 | 66.66±1.11 | 74.05±2.16 | 73.55±2.01 |
| **TS2Vec(2.58M)** | 78.57±0.00 | 78.46±0.00 | 84.29±2.86 | 84.26±2.90 | 89.23±5.65 | 89.09±5.79 | 72.22±3.51 | 71.92±3.75 | 81.08±4.52 | 80.70±5.12 |
| **BIOT(4.16M)** | 85.71±0.00 | 84.44±0.00 | 64.29±6.39 | 63.38±6.28 | 88.46±0.00 | 88.33±0.06 | 73.33±3.33 | 73.14±3.45 | 73.51±4.95 | 72.53±5.95 |
| **EEG2Rep(5.33M)** | 75.71±5.71 | 75.59±5.74 | 78.57±6.39 | 78.07±6.59 | 89.23±2.88 | 89.21±2.87 | 76.67±4.84 | 76.61±4.90 | 82.70±2.76 | 82.55±2.89 |
| **LaBraM(9.62M)** | 57.14±0.00 | 72.73±0.00 | 50.00±0.00 | 66.67±0.00 | 69.23±5.44 | 69.69±5.43 | 72.78±3.69 | 73.56±3.18 | 65.41±2.02 | 67.80±3.23 |
| **EEGPT(25.5M)** | 57.14±0.00 | 72.73±0.00 | 50.00±0.00 | 66.67±0.00 | 60.00±5.76 | 61.08±4.96 | 62.78±7.37 | 68.54±3.98 | 61.08±3.67 | 67.64±3.81 |
| **LEAD-Vanilla(3.21M)** | 82.86±3.50 | 82.81±3.55 | 75.71±5.71 | 75.39±5.78 | 94.62±1.88 | 94.59±1.90 | 73.33±2.22 | 73.27±2.21 | 73.51±4.32 | 72.72±4.71 |
| **LEAD-Sup(3.21M)** | **91.43±2.86** | **91.34±2.81** | 78.57±0.00 | 78.46±0.00 | 95.38±1.54 | 95.38±1.54 | 77.78±1.76 | 77.71±1.81 | 80.54±2.02 | 80.42±2.04 |
| **LEAD-Base(3.41M)** | 80.00±5.35 | 79.96±5.36 | **90.00±3.50** | **89.98±3.48** | **100.00±0.00** | **100.00±0.00** | **84.44±2.22** | **84.42±2.21** | **91.89±1.71** | **91.86±1.73** |

comparison between LEAD-Sup and LEAD-Base indicates that proper self-supervised pre-training methods dramatically reduce the interference of subject features and improve inter-subject classification ability. Besides, We observe that subject-level classification results are typically better than sample-level classification results for almost all methods. This demonstrates that majority voting does alleviate noise interference from outlier samples within a subject. The improvement is particularly notable in the two Cognision datasets. The best performance for these two datasets on the sample-level is around 70% F1 score, but increases to approximately 90% with majority voting. Since these datasets were collected in an industrial pipeline with a balanced number of samples per subject (300 or 400), we can infer that the more balanced the number of samples per subject, the greater the improvement introduced by majority voting.

### 3.2. Ablation Study and Supplementary Experiments

We conduct comprehensive ablation studies, including the effectiveness of non-AD and AD datasets, contrastive learning modules research, and training setups, see Appendix F. Besides, we conduct additional experiments for brain interpretability analysis, including frequency bands analysis and channels analysis. See Appendix G for more details.

## 4. Conclusion

In this paper, we present **LEAD**, the world's first large foundational model for EEG-Based Alzheimer's Disease detection on the world's largest EEG-based AD detection corpus, including 813 subjects. We design a complete pipeline encompassing dataset selection, data processing, pre-training framework, model architecture, and evaluation metrics. We perform self-supervised pre-training on 4 AD datasets and 7 non-AD neurological diseases or healthy control datasets, totaling 2,354 subjects. The self-supervised pre-training includes sample-level and subject-level contrastive learning. Unified fine-tuning is performed on 5 AD datasets with channel-aligned datasets, totaling 615 subjects. We use a backbone encoder that can leverage both temporal and spatial features. The significant improvement compared with state-of-the-art baselines demonstrates the effectiveness of our design for dataset selection, channel alignment, self-supervised pre-training, and unified fine-tuning. We hope to inspire future research on EEG-based AD detection and other neurological disease detection, such as Parkinson's disease. More discussion on the existing large EEG model, the effectiveness of our subject-level contrasting, and limitations & future works are presented in Appendix H.

## Impact Statement

This paper introduces the first large foundational model for EEG-based Alzheimer's Disease detection, trained on the largest EEG-AD corpus to date. Our results demonstrate the effectiveness of large pre-trained models and multi-dataset fine-tuning for AD detection, provided that appropriate training methods and datasets are selected. Our approach significantly outperforms methods trained on single datasets and other state-of-the-art self-supervised pre-training methods and large EEG foundation models trained on multiple datasets. The subject-independent evaluation, which tests on unseen subjects, further highlights the applicability of our method in real-world scenarios. We open-source our code, pre-trained model, and fine-tuned model with the hope that this work will drive progress in EEG-based AD detection and inspire future research in detecting other brain disorders and neurodegenerative diseases.

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

# A. Related Work

## A.1. EEG-Based Alzheimer's Disease Detection

In the last two decades, EEG-based Alzheimer's disease (AD) detection has followed two main research directions: manual biomarker extraction and deep learning representation. **Biomarker Extraction:** This research direction aims to identify potential biomarkers in EEG signals of AD patients and use simple classifiers, such as Multi-Layer Perceptrons (MLP) and Support Vector Machines (SVM), to differentiate these features from normal healthy subjects. Different types of EEG features are used, including statistical features like Mean, Skewness, Kurtosis, and Standard Deviation (Tzimourta et al., 2019b;a; Kulkarni & Bairagi, 2017; Kanda et al., 2014; Waser et al., 2013; Tylova et al., 2013; Mora-Sánchez et al., 2019), spectral features like Phase Shift, Phase Coherence, Bispectrum, and Bicoherence (Wang et al., 2017; Cassani et al., 2014; Wang et al., 2015; Fraga et al., 2013; Tait et al., 2019; Waser et al., 2016; Trambaiolli et al., 2011), power features like Power Spectrum Density, Relative Band Power, Ratio of EEG Rhythm, and Energy (Fahimi et al., 2017; Schmidt et al., 2013; Liu et al., 2016; Kanda et al., 2014), as well as complexity features like Shannon Entropy, Tsallis Entropy, and Permutation Entropy (Garn et al., 2015; Azami et al., 2019; Tylová et al., 2018; Coronel et al., 2017; Al-Nuaimi et al., 2018). The main advantage of this approach is its interpretability, which is crucial for real-world healthcare applications. **Deep Learning:** Compared to manual biomarker extraction, deep learning offers an alternative approach by automatically extracting useful representations for AD detection. Models such as Convolutional Neural Networks (CNNs) (Li et al., 2022; Cura et al., 2022), Graph Neural Networks (GNNs) (Shan et al., 2022; Klepl et al., 2023), and Transformers (Wang et al., 2024e) are widely used for representation learning. Some researchers still perform manual feature extraction or transform the data before applying deep learning models. For example, the method in (Ieracitano et al., 2019a) converts 5-second EEG intervals into Power Spectral Density (PSD) images and uses 2D convolutional layers on the images for feature extraction. DICE-net (Miltiadous et al., 2023a) extracts relative band power and spectral coherence connectivity across five frequency bands and applies convolutional layers followed by transformers. In contrast, some studies apply deep learning methods directly to EEG data. For instance, the method in (Gallego-Viñarás et al., 2024) uses semi-supervised spatiotemporal representation learning with deep learning models for AD detection based on different sleep-stage EEG data. STEADYNet (Kachare et al., 2024) designs low-complexity convolutional models for AD and dementia detection, focusing on fast inference times. Research in (Watanabe et al., 2024) using MNet that applies convolutional networks for feature extraction and concatenates with relative power spectrum for AD and other dementia detection. ADformer (Wang et al., 2024e) uses a multi-granularity spatial-temporal transformer for AD detection and widely tests on five EEG-AD datasets.

## A.2. Self-Supervised Learning in EEG

There are two main strategies for self-supervised representation learning in EEG: contrastive learning and mask-reconstruction. **Contrastive Learning:** BENDR (Kostas et al., 2021) follows a similar contrastive learning pipeline as Wav2Vec (Baevski et al., 2020), but it is trained on EEG data. EEG2Vec (Zhu et al., 2023) explores both contrastive learning and mask-reconstruction for self-supervised pre-training on EEG data. BIOT (Yang et al., 2024) designs a transformer architecture for biomedical signal embedding and applies a self-supervised contrastive framework similar to BYOL (Grill et al., 2020). COMET (Wang et al., 2024b) utilizes various data levels in biomedical time series to define positive and negative pairs in contrastive learning. **Mask-Reconstruction:** Neuro-BERT (Wu et al., 2024) employs masked autoencoding to predict missing amplitude and phase of EEG signals during pre-training. EEG2Rep (Mohammadi Foumani et al., 2024) combines a context encoder with a momentum target encoder to reconstruct context-level representations rather than raw data in self-supervised pre-training. LaBraM (Jiang et al., 2024), the first large foundation model in the EEG domain, uses a neural tokenizer to reconstruct the Fourier spectrum during self-supervised pre-training. EEGPT (Wang et al., 2024a) is a foundation model for EEG representation learning that integrates reconstruction loss with an alignment loss between the encoder and momentum encoder. **Other strategies:** Recently, some work has begun exploring the potential of autoregressive pretraining for EEG, such as another work also named EEGPT (Yue et al., 2024).

# B. Indices Shuffling Algorithm

To avoid overlapping subject IDs when loading data from multiple datasets, we first count the number of subjects in each dataset and assign each subject a unique subject ID starting from 1. As a result, each sample $x_i$ has a corresponding new subject ID $s_i$, where $s_i = s_j$ indicates that $x_i$ and $x_j$ are from the same subject. In real-world scenarios, as the number of subjects increases, the likelihood of samples with the same subject ID appearing in the same training batch decreases. This situation may hinder subject-level contrastive learning.

**Algorithm 1** Pseudo code of Indices Shuffling.

```
import numpy as np
# ids: subject IDs array
# bs: pseudo batch size for shuffling
# gs: pseudo group size for shuffling

def shuffle_indices(ids, batch_size=128, group_size=2)
    # indices sorted by subject IDs
    indices = np.argsort(subject_ids)
    length = len(indices)

    # split indices into groups
    groups = [indices[i:i + group_size] for i in range(0, length, group_size)]

    # shuffle groups
    np.random.shuffle(groups)
    # concatenate groups
    indices = np.concatenate(groups)

    # split indices into batches
    batches = [indices[i:i + batch_size] for i in range(0, length, batch_size)]

    # shuffle indices in the batch
    for batch in batches:
        np.random.shuffle(batch)
    # concatenate batches
    indices = np.concatenate(batches)

    return indices
```

To address this issue, we develop an indices shuffling algorithm, which is called every epoch by passing it to the *sampler* parameter in the PyTorch *DataLoader*. We first sort the indices by subject IDS and split the sorted indices of the entire training set into small groups (where the group size is much smaller than the batch size), each containing indices of samples from the same subject ID. We then randomly shuffle these groups rather than shuffle the individual samples. After shuffling the groups, we split the shuffled indices into batches and shuffle the indices within each batch. This two-step shuffling process ensures the randomness of the samples in each training epoch while maintaining a relatively balanced number of positive pairs for subject-level contrastive learning. The pseudocode for indices shuffling is provided in Algorithm 1.

## C. Data Augmentation Banks

We apply data augmentation for self-supervised contrastive pretraining and some supervised learning methods. We utilize a bank of data augmentation techniques to enhance the model's robustness and generalization. During the forward pass in the training of each iteration, one augmentation method will be picked from available augmentation options with equal probability. The data augmentation methods include temporal flipping, temporal masking, frequency masking, channel masking, jittering, and dropout, and can be further expanded to more choices. We provide a detailed description of each technique below.

**Temporal Flippling.** We reverse the EEG data along the temporal dimension. The probability of applying this augmentation is controlled by a parameter *prob*, with a default value of 0.5.

**Temporal Masking.** We randomly mask timestamps across all channels. The proportion of timestamps masked is controlled by the parameter *ratio*, with a default value of 0.1.

**Frequency Masking.** First introduced in (Zhang et al., 2022) for contrastive learning, this method involves converting the EEG data into the frequency domain, randomly masking some frequency bands, and then converting it back. The proportion of frequency bands masked is controlled by the parameter *ratio*, with a default value of 0.1.

**Channel Masking.** We randomly mask channels across all timestamps. The proportion of channel masked is controlled by the parameter *ratio*, with a default value of 0.1.

**Jittering.** Random noise, ranging from 0 to 1, is added to the raw data. The intensity of the noise is adjusted by the parameter *scale*, which is set by default to 0.1.

**Dropout.** Similar to the dropout layer in neural networks, this method randomly drops some values. The proportion of values dropped is controlled by the parameter *ratio*, with a default value of 0.1.

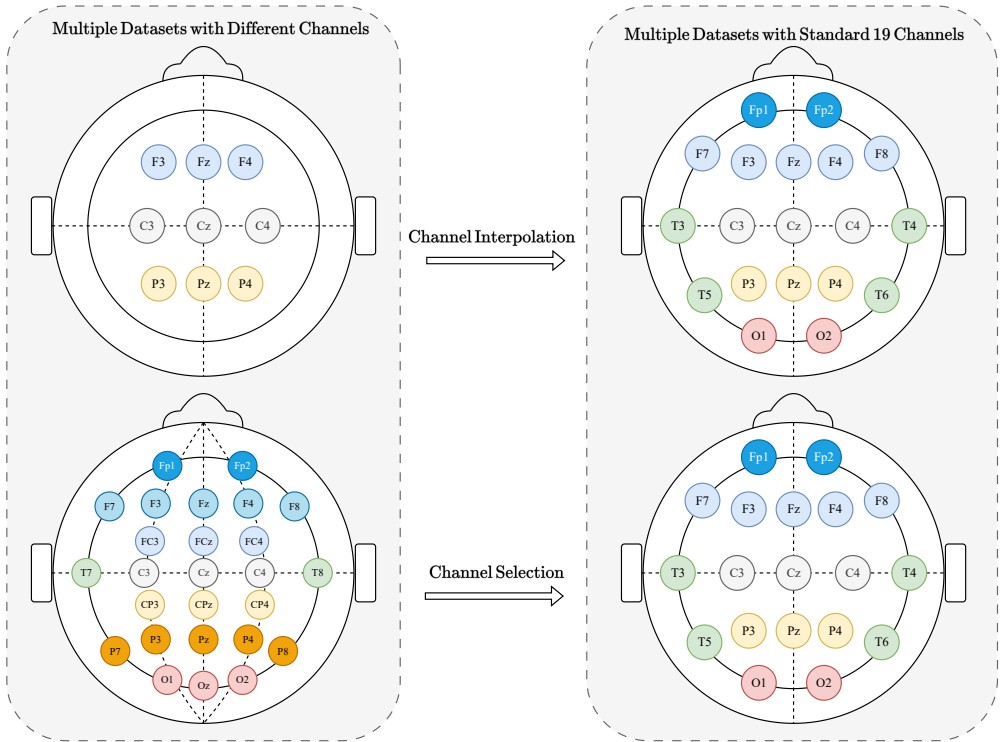

*Figure 3.* **Channel Alignment.** We perform channel alignment to ensure that all datasets have the standard 19 channels according to the 10-20 international system. For datasets with more than 19 channels, we select the 19 channels based on their names. For datasets with fewer than 19 channels, we perform channel interpolation. In cases of channel name mismatches, we select the closest channels as alternatives by calculating their 3D coordinates.

## D. Datasets Preprocessing

We refer to datasets that include Alzheimer's Disease (AD) subjects as AD datasets, while datasets that do not include AD subjects are called non-AD datasets. In total, we have 9 AD datasets and 7 non-AD datasets. Note that all the non-AD datasets have one commonality: the label is assigned to each subject, which adapts to the subject-level contrastive learning. Among the AD datasets, **ADFTD**, **BrainLat**, **CNBPM**, **Cognision-ERP**, and **Cognision-rsEEG** are selected for downstream evaluations due to their high quality, larger number of subjects, and sufficiently long recording trials per subject, which provide a more robust assessment. The remaining 4 AD datasets, as well as all the non-AD datasets, are used for self-supervised pretraining to learn general EEG patterns and disease-specific patterns related to neurological diseases.

For datasets where the raw channel names or numbers do not match the 19 standard channels (Fp1, Fp2, F7, F3, Fz, F4, F8, T3, C3, Cz, C4, T4, T5, P3, Pz, P4, T6, O1, O2) in the 10-20 international system, we perform channel alignment to generate two versions of the processed data: a channel-aligned version with 19 channels and a raw-channel version with either more or fewer channels. The channel-aligned version is used for pretraining and unified fine-tuning across multiple datasets, while the raw-channel version without channel alignment is used for supervised learning on individual datasets. These two versions aim to demonstrate the effectiveness of the pretraining and unified fine-tuning method pipeline compared to supervised learning on individual datasets, even if it involves a trade-off where some channel information is lost in certain datasets. Note that the channels T7, T8, P7, and P8 are the same as the channels T3, T4, T5, and T6 in the international 10-20 and 10-10 system (Acharya et al., 2016). Figure 3 illustrates the channel alignment process. The statistics of the processed datasets are presented in Table 1.

There are three main reasons for aligning all datasets to the same 19 channels. First, these 19 channels are the most commonly used in EEG-based AD detection (Aviles et al., 2024), matching the goal of low-cost and convenient AD detection through EEG. Second, the standard 19 channels cover all the brain regions, preserving enough temporal and spatial information. This channel alignment approach avoids the trade-off between computational resources and patch length in existing methods that seek to capture both spatial and fine-grained temporal features (Yang et al., 2024; Wang et al., 2024a; Jiang et al., 2024), as we discuss later in Appendix H. Third, using the same channels across datasets enables unified fine-tuning for downstream

AD datasets together in one run, which significantly improves performance compared to fine-tuning on individual datasets, a benefit we demonstrate later in the ablation study 8.

The final processed datasets are organized into two folders: *Feature/* and *Label/*. The *Feature/* folder contains files named in the format *feature_ID.npy* for all subjects, where *ID* represents the subject ID. Each *feature_ID.npy* file contains samples belonging to the same subject, stacked into a 3-D array with the shape $[N_{sample}, T, C]$, where $N_{sample}$ denotes the number of samples in this subject, $T$ denotes the number of timestamps per sample, and $C$ denotes the number of channels. Note that different subjects may have different numbers of samples. The *Label/* folder contains a file named *label.npy*, which is a 2-D array with the shape $[N_{subject}, 2]$, where $N_{subject}$ is the total number of subjects. The first column contains the subject's label (e.g., healthy or AD), and the second column contains the subject ID, which ranges from 1 to $N_{subject}$.

### D.1. AD Datasets

#### D.1.1. AD-AUDITORY.

The AD-Auditory (40Hz Auditory Entrainment) is a publicly available EEG dataset on the OpenNEURO website[8] from the paper (Lahijanian et al., 2024). It contains 35 subjects, including 17 AD, 6 MCI, 10 healthy controls, and 2 unknown subjects. This dataset aims to investigate the effect of entrainment on brain oscillations using EEG signal recordings during auditory brain stimulation for distinguish Alzheimer's Disease. All the data are recorded using 19 monopolar channels (Fp1, Fp2, F7, F3, Fz, F4, F8, T3, C3, Cz, C4, T4, T5, P3, Pz, P4, T6, O1, and O2) based on the standard 10/20 system, with a sampling rate set to 250Hz. The dataset's authors preprocess the data using the EEGLab toolbox in Matlab, which includes bandpass filtering, noise removal, artifact removal, re-referencing, and interpolating rejected channels, as described in their paper and on the data website. We perform secondary data preprocessing to match the pipeline of our method.

Each subject has one recording trial. For each raw trial, we first downsample the trials from 250Hz to 128Hz. Then, we segment all the trials into 1-second samples with 128 timestamps. We drop the last sample if it is shorter than 128 timestamps. This results in a total of 37,425 1-second, 128Hz samples. We apply bandpass filtering ranging from 0.5Hz to 45Hz, followed by standard normalization on each channel. We perform preliminary evaluations on this dataset and find substantial variability among subjects. We suspect the limited number of subjects and potential data and label quality issues cause this variability. As a result, we decide to use this dataset for pre-training, although it is an AD dataset.

#### D.1.2. ADFSU.

This is a publicly available dataset provided by Dr. Dennis Duke of Florida State University (Vicchietti et al., 2023; Nour et al., 2024), as we name it to ADFSU[9]. It contains data from 80 AD subjects and 12 healthy subjects. Each subject has a recording with a sampling frequency of 128Hz and an 8-second trial collected across 19 standard channels (Fp1, Fp2, F7, F3, Fz, F4, F8, T3, C3, Cz, C4, T4, T5, P3, Pz, P4, T6, O1, and O2) during both eye-open and eye-closed resting-state conditions. The preprocessing steps includes band-pass filtering within the range of 0.5–30 Hz, and an experienced EEG expert removes artifacts caused by movements. We perform secondary preprocessing to match the pipeline of our method.

The data files are organized into "AD" and "Healthy" folders, with each folder containing "Eyes_open" and "Eyes_closed" subfolders to indicate the different tasks. Each task folder contains subject folders labeled with ID numbers, and all the data are stored in channel_name.txt files. Note that the eye-open data for the healthy subject with ID 5 is empty. For simplicity, we manually copy the eyes-closed data for this subject and use it as the eye-open data to avoid handling empty files in the code. For each subject, we first load all the channel text files and concatenate them into two recording trials: eyes-open and eyes-closed. We segment the data for each trial into 1-second, half-overlapping samples with 128 timestamps. We discard the last sample if it is shorter than 128 timestamps. This results in a total of 2,760 1-second, 128Hz samples. We apply bandpass filtering in the range of 0.5Hz to 45Hz, followed by standard normalization on each channel. Due to the extreme imbalance between AD and healthy subjects, and the limited length of each subject's recording trial, we use this dataset for pretraining rather than downstream evaluation.

---

[8]https://openneuro.org/datasets/ds005048/versions/1.0.0

[9]https://osf.io/2v5md/

### D.1.3. ADFTD.

The ADFTD (A dataset of EEG recordings from Alzheimer's disease, Frontotemporal dementia and Healthy subjects) is a publicly available EEG dataset on the OpenNEURO website[10] from the paper (Miltiadous et al., 2023b;a). It contains 88 subjects, including 36 AD, 23 Frontotemporal Dementia (FTD), and 29 healthy controls. For recording, a Nihon Kohden EEG 2100 clinical device is used, with 19 scalp electrodes (Fp1, Fp2, F7, F3, Fz, F4, F8, T3, C3, Cz, C4, T4, T5, P3, Pz, P4, T6, O1, and O2) according to the 10-20 international system and 2 reference electrodes (A1 and A2) placed on the mastoids for impedance check, according to the manual of the device. Each recording is performed according to the clinical protocol, with participants sitting with their eyes closed. The collection sampling rate is 500Hz. The dataset's authors preprocess the data using the EEGLab toolbox in Matlab, which includes bandpass filtering, noise removal, artifact removal, re-referencing, and interpolating rejected channels, as described in their paper and on the data website. We perform secondary data preprocessing to match the pipeline of our method.

Each subject has one recording trial. This paper only uses 65 subjects that are AD and healthy control subjects. For each raw trial, we first downsample the trials from 500Hz to 128Hz. Then, we segment all the trials into 1-second samples with 128 timestamps. We drop the last sample if it is shorter than 128 timestamps. This results in a total of 53,215 1-second, 128Hz samples. We apply bandpass filtering ranging from 0.5Hz to 45Hz, followed by standard normalization on each channel. This AD dataset is used for downstream evaluation as it contains enough subjects, and each subject has a long enough recording to be segmented into samples.

### D.1.4. ADSZ.

The ADSZ (Alzheimer's Disease and Schizophrenia) dataset is a public EEG dataset[11] from the paper (Alves et al., 2022; Pineda et al., 2020). We use only the sub-dataset for Alzheimer's disease (AD) available in the download link. This dataset contains data from 48 subjects, including 24 AD subjects and 24 healthy elderly subjects. The data are collected from 19 standard channels (Fp1, Fp2, F7, F3, Fz, F4, F8, T3, C3, Cz, C4, T4, T5, P3, Pz, P4, T6, O1, and O2) during eyes-open and eyes-closed resting states, with a sampling frequency of 128Hz. Most subjects have an EEG trial duration of 8 seconds, although some trials last 10, 12, or 14 seconds, with timestamps ranging from 1,024 to 1,792. The preprocessing of the signals includes band-pass filtering within the range of 1–30 Hz, and an experienced EEG technician removes artifacts caused by subject movements. We perform secondary preprocessing to match the pipeline of our method.

For each trial, we segment the data into 1-second, half-overlapping samples with 128 timestamps. We discard the last sample if it is shorter than 128 timestamps. This results in a total of 768 1-second, 128Hz samples. We apply bandpass filtering in the range of 0.5Hz to 45Hz, followed by standard normalization on each channel. Due to the limited number of subjects and the short duration of each subject's recording trials, we use this dataset for pretraining rather than downstream evaluation.

### D.1.5. APAVA.

The APAVA (Alzheimer's Patients' Relatives Association of Valladolid) dataset[12], referenced in the study by (Escudero et al., 2006), is a publicly available EEG dataset consisting of 23 subjects, including 12 AD subjects and 11 healthy elderly subjects. The data are recorded using 16 channels (Fp1, Fp2, F7, F3, F4, F8, T3, C3, C4, T4, T5, P3, P4, T6, O1, and O2) with a sampling frequency of 256Hz. Each subject has multiple trials, with each trial lasting 5 seconds, corresponding to 1,280 timestamps. A specialist physician visually inspects the recordings to select data with minimal movement, electromyographic activity, or electrooculographic artifacts. A bandpass filter is applied with 0.5 Hz and 40 Hz cut-off frequencies. We perform secondary preprocessing to match the pipeline of our method.

Since this dataset has only 16 channels compared to the 19 standard channels, we perform channel alignment using the Python MNE EEG processing tools. Specifically, by checking the file information stored in the Matlab files, we found that the three missing channels are Fz, Cz, and Pz. We set these channels as "bad channels" and interpolate them using the montage "standard_1020." After this, we have recording trials in all 19 standard channels. We then downsample all the trials from 256Hz to 128Hz. We segment the data into 1-second, half-overlapping samples for each trial with 128 timestamps. We discard the last sample if it is shorter than 128 timestamps. This results in a total of 5,967 1-second, 128Hz samples. We apply bandpass filtering in the range of 0.5Hz to 45Hz, followed by standard normalization on each channel. Besides, we

---

[10]https://openneuro.org/datasets/ds004504/versions/1.0.8
[11]https://figshare.com/articles/dataset/Alzheimer_s_disease_and_Schizophrenia/19091771
[12]https://osf.io/jbysn/

also process a raw 16-channel data version for supervised learning on individual datasets by not doing channel alignment and keeping other procedures the same. Due to the limited number of subjects and the short duration of each subject's recording trials, we use this dataset for pretraining rather than downstream evaluation.

### D.1.6. BRAINLAT.

The BrainLat[13] (Latin American Brain Health Institute) dataset comprises multimodal neuroimaging data from 780 participants from Latin America (Prado et al., 2023). It contains two modalities: EEG and MRI. It includes five classes of subjects: Alzheimer's disease (AD), behavioral variant frontotemporal dementia (bvFTD), multiple sclerosis (MS), Parkinson's disease (PD), and healthy controls (HC). For EEG data recording, subjects are recorded in an eye-closed resting state inside a dimly lit, sound-attenuated, and electromagnetically shielded EEG room. They are instructed to remain still and awake, with a 128-channel Biosemi Active-two acquisition system (pin-type, active, sintered Ag-AgCl electrodes). The data are band-pass filtered between 0.5 and 40 Hz using a zero-phase shift Butterworth filter of order 8. The data are then downsampled to 512 Hz, and Independent Component Analysis (ICA) is used to correct EEG artifacts induced by blinking and eye movements. We perform secondary preprocessing to match the pipeline of our method.

In this paper, we use only the EEG data from the AD and HC classes for research. EEG data for each subject are stored in folders labeled AR and CL, representing the subjects' countries: Argentina and Chile. It is important to note that some subjects cannot read for unknown reasons, such as the subject named "sub-100013" (at least when we downloaded the dataset, which the data version was last modified by Dr. Pavel Prado on 7/2/2024). Additionally, not all subjects have EEG data; most subjects only have MRI datasets. In total, there are 135 functional subjects with EEG data across all five classes, with 67 subjects(35 AD and 32 HC) used for this paper.

Since this dataset uses the Biosemi128 montage instead of the standard_1020 montage, which has an entirely different electrode naming and positioning scheme, we perform channel alignment using the Python MNE EEG processing tools. Specifically, we use the 3-D coordinates of the channels to identify 19 channels in the Biosemi128 montage closest to the 19 standard channels in the 10-20 system. The closest channels are C29, C16, D7, D4, C21, C4, C7, D24, D19, A1, B22, B14, A10, A18, A19, B4, B7, A16, and A29, which we use as replacements for the 19 standard channels. We then downsample all trials from 512Hz to 128Hz. We segment the data into 1-second, 128Hz samples with 128 timestamps. We discard the last sample if it is shorter than 128 timestamps. This results in a total of 29,788 1-second, 128Hz samples. We apply bandpass filtering from 0.5Hz to 45Hz, followed by standard normalization on each channel. We also process a raw 128-channel data version for supervised learning on individual datasets by not performing channel alignment while keeping all other procedures. This high-quality dataset has enough subjects and trial recording length; we use it for downstream evaluations.

### D.1.7. CNBPM.

The CNBPM is a large private EEG dataset provided by the AI-LAB laboratory at the University Mediterranea of Reggio Calabria, Italy, referenced in studies (Ieracitano et al., 2019b; Amezquita-Sanchez et al., 2019). It consists of 63 subjects with Alzheimer's Disease (AD), 63 with Mild Cognitive Impairment (MCI), and 63 Healthy Control (HC) subjects. The data are collected using 19 standard channels (Fp1, Fp2, F7, F3, Fz, F4, F8, T3, C3, Cz, C4, T4, T5, P3, Pz, P4, T6, O1, and O2) with an initial sampling rate of 1024Hz. A frequency-band filter is applied to filter the frequency bands between 0.5 and 32 Hz, followed by downsampling to reduce the sampling rate to 256Hz. Visible blinks affected by artifacts are visually inspected and removed by an EEG expert. We perform secondary preprocessing to match the pipeline of our method.

In this paper, we use only the EEG data of 63 AD and 63 HC for research. For each subject's recording trial, which ranges from several minutes to over half an hour, we first downsample all trials from 256Hz to 128Hz. We then segment the trials into 1-second, non-overlapping samples, excluding those shorter than 1 second at the trial's edge. This results in a total of 46,336 1-second, 128Hz samples. We apply bandpass filtering ranging from 0.5Hz to 45Hz, followed by standard normalization on each channel. Since this dataset is sufficiently large and high-quality, with more than 100 subjects and long recording trials per subject, we use it for downstream evaluations.

---

[13]https://www.synapse.org/Synapse:syn51549340/wiki/624187

D.1.8. COGNISION-ERP.

The Cognision-ERP is a private Event-Related Potential (ERP) EEG dataset from the *Cognision*[14] company, referenced in the study (Cecchi et al., 2015). It contains 177 subjects, including 90 Alzheimer's Disease (AD) subjects and 87 Healthy Control (HC) subjects. The total number of samples is 61,300, with each subject having either 300 or 400 samples, and each sample containing 149 timestamps. The sampling rate is 125Hz, and there are 7 channels (Fz, Cz, Pz, F3, P3, F4, P4). Artifacts, such as eye movements, are visually inspected and removed by an EEG expert. We perform secondary preprocessing to match the pipeline of our method.

Since this dataset has only 7 channels compared to the 19 standard channels, we perform channel alignment using the Python MNE EEG processing tools. Specifically, the 12 missing channels are Fp1, Fp2, F7, F8, T3, C3, C4, T4, T5, T6, O1, and O2. We set these channels as "bad channels" and interpolate them using the montage "standard_1020." After this, we have recording samples with all 19 standard channels. We then upsample all the samples from 125Hz to 128Hz, which increases the number of timestamps per sample from 149 to 153. We take the middle 128 timestamps as the new sample. This results in a total of 61,300 1-second, 128Hz samples. We apply bandpass filtering in the range of 0.5Hz to 45Hz, followed by standard normalization on each channel. We also process a raw 7-channel data version for supervised learning on individual datasets by not performing channel alignment while keeping all other procedures the same. Since this dataset is sufficiently large and high-quality, with more than 100 subjects and many samples per subject, we use it for downstream evaluations.

D.1.9. COGNISION-RSEEG.

The Cognision-rsEEG is a private EEG dataset from the *Cognision* company. Unlike Cognision-ERP, Cognision-rsEEG consists of resting-state EEG data. It contains 180 subjects, including 97 Alzheimer's Disease (AD) subjects and 83 Healthy Control (HC) subjects. Each subject has a recording trial with 22,524 timestamps collected at 125Hz. The number of channels is 7 (Fz, Cz, Pz, F3, P3, F4, P4). Artifacts, such as eye movements, are visually inspected and removed by an EEG expert. We perform secondary preprocessing to match the pipeline of our method.

Since this dataset has only 7 channels compared to the 19 standard channels, we perform channel alignment using the Python MNE EEG processing tools. Specifically, the 12 missing channels are Fp1, Fp2, F7, F8, T3, C3, C4, T4, T5, T6, O1, and O2. We set these channels as "bad channels" and interpolate them using the "standard_1020" montage. After this, we obtain recording samples with all 19 standard channels. We then upsample all the samples from 125Hz to 128Hz. Next, we segment the trials into 1-second, non-overlapping samples, excluding those shorter than 1 second at the edges of the trial. This results in a total of 32,400 1-second, 128Hz samples. We apply bandpass filtering in the range of 0.5Hz to 45Hz, followed by standard normalization on each channel. Since this dataset is sufficiently large and high-quality, with more than 100 subjects and many samples per subject, we use it for downstream evaluations.

**D.2. Non-AD Datasets**

D.2.1. DEPRESSION.

The Depression (EEG: Depression rest) dataset is a publicly available EEG dataset on the OpenNEURO website[15] from the paper (Cavanagh et al., 2019; Cavanagh, 2021). It contains data from 122 college-age subjects with healthy and different degrees of depression. The EEG data are recorded in a resting state, with instructions for eyes open and eyes closed, triggering one-minute spans of either open or closed eyes. Each subject's depression level is labeled based on their score on the Beck Depression Inventory (BDI). The raw data sampling frequency is 500Hz. We perform secondary preprocessing to match the pipeline of our method.

Each subject has one or multiple recording trials. We check the channel information for each trial and find that some trials have 66 channels and others have 67 channels. We perform channel alignment by selecting the 19 standard channels: Fp1, Fp2, F7, F3, Fz, F4, F8, T7, C3, Cz, C4, T8, P7, P3, Pz, P4, P8, O1, and O2. Note that the channels T7, T8, P7, and P8 are the same as the channels T3, T4, T5, and T6 in the international 10-20 and 10-10 system (Acharya et al., 2016). After alignment, we obtain recording trials with all 19 standard channels. We then downsample all the trials from 500Hz to 128Hz. Next, we segment the trials into 1-second, non-overlapping samples, excluding those shorter than 1 second at the edges of the trial. This results in a total of 24,014 1-second, 128Hz samples. This 19-channel processed dataset is used for

---

[14]https://www.cognision.com/
[15]https://openneuro.org/datasets/ds003478/versions/1.1.0

self-supervised contrastive pretraining. We apply bandpass filtering in the range of 0.5Hz to 45Hz, followed by standard normalization on each channel. We also process a raw 66-channel data version for supervised learning on individual datasets by not performing channel alignment while keeping all other procedures the same. The 66 channels are picked by the most common channel names among trials.

### D.2.2. PEARL-NEURO.

The PEARL-Neuro (A Polish Electroencephalography, Alzheimer's Risk-genes, Lifestyle and Neuroimaging) dataset is a publicly available EEG dataset on the OpenNEURO website[16], referenced in the paper (Dzianok & Kublik, 2024). The full dataset contains data from 192 self-reported healthy middle-aged (50-63) subjects, with a balanced female-to-male ratio. Of these, 79 subjects are publicly available, and the dataset includes two modalities: EEG and fMRI. Other information, such as blood tests, demographics, and other health conditions, are also provided. The dataset aims to identify genetic variations associated with brain anatomical and functional phenotype imaging genomics, which could be potential biomarkers for predicting the risk of developing neurological and psychiatric disorders. This could lead to earlier diagnoses, more targeted treatments, and improved patient outcomes. EEG data are recorded using Brain Products systems, including an actiCHamp amplifier and high-density actiCAP electrode caps with 128 electrodes (Brain Products GmbH, Munich, Germany). The FCz electrode is used as an online reference, and the sampling rate is set to 1000Hz with a low-pass filter at 280Hz. The dataset includes three different tasks: the Sternberg memory task (Sternberg), the Multi-source interference task (MSIT), and resting-state (rest). In this paper, we use only the resting-state EEG data. We perform secondary preprocessing to match the pipeline of our method.

For resting-state trials from each subject, we first align the channels by selecting 19 standard channels: Fp1, Fp2, F7, F3, Fz, F4, F8, T7, C3, Cz, C4, T8, P7, P3, Pz, P4, P8, O1, and O2, where T7, T8, P7, and P8 correspond to the channels T3, T4, T5, and T6 in the international 10-20 system. After alignment, we obtain recording trials with all 19 standard channels. We then downsample all trials from 1000Hz to 128Hz. Next, we segment the trials into 1-second, non-overlapping samples, excluding those shorter than 1 second at the edges of the trial. This results in a total of 51,670 1-second, 128Hz samples. We apply bandpass filtering in the range of 0.5Hz to 45Hz, followed by standard normalization on each channel. This 19-channel processed dataset is used for self-supervised contrastive pretraining. Additionally, we process a 127-channel data version for supervised learning on individual datasets without channel alignment but keeping all other procedures the same.

### D.2.3. REEG-BACA.

The REEG-BACA (Resting-state EEG data before and after cognitive activity across the adult lifespan and a 5-year follow-up) dataset is a publicly available EEG dataset on the OpenNEURO website[17], referenced in the paper (Getzmann et al., 2024). According to the paper's description, this dataset consists of 64 channels based on the 10-20 system, with the FCz electrode as an online reference. It includes resting-state EEG recordings from 608 subjects aged between 20 and 70 years, along with follow-up measurements of 208 subjects approximately 5 years later, starting in 2021. The EEG data are recorded with eyes open and eyes closed before and after a 2-hour block of cognitive experimental tasks. The EEG data are recorded at a 1000Hz sampling rate and filtered online using a 250Hz low-pass filter. This dataset aims to study the aging of brain activity in a resting state and provide a normal distribution of healthy subjects' resting-state EEG for comparison with clinically relevant disorders. We perform secondary preprocessing to match the pipeline of our method.

For resting-state trials in both the eye-open and eye-closed conditions from each subject, we first align the channels by selecting the 19 standard channels: Fp1, Fp2, F7, F3, Fz, F4, F8, T7, C3, Cz, C4, T8, P7, P3, Pz, P4, P8, O1, and O2. Note that T7, T8, P7, and P8 are the same channels as T3, T4, T5, and T6 in the international 10-20 and 10-10 systems. After alignment, we obtain recording trials with all 19 standard channels. We then downsample all trials from 1000Hz to 128Hz. Next, we segment the trials into 1-second, non-overlapping samples, excluding those shorter than 1 second at the edges of the trial. This results in a total of 611,269 1-second, 128Hz samples. We apply bandpass filtering in the range of 0.5Hz to 45Hz, followed by standard normalization on each channel. This 19-channel processed dataset is used for self-supervised contrastive pretraining. Additionally, we process a 65-channel data version for supervised learning on individual datasets by not performing channel alignment while keeping all other procedures the same.

---

[16]https://openneuro.org/datasets/ds004796/versions/1.0.9
[17]https://openneuro.org/datasets/ds005385/versions/1.0.2

### D.2.4. REEG-PD.

The REEG-PD (Rest eyes open) dataset is a publicly available EEG dataset on the OpenNEURO website[18], referenced in the paper (Singh et al., 2023). This dataset includes 149 subjects, with 100 Parkinson's disease (PD) subjects and 49 Healthy controls (HC) subjects. According toe the description in their paper, the EEG data is recorded with a 64-channel BrainVision cap in a resting state with their eyes open for two minutes. The sampling frequency is set to 500Hz, and a 0.1Hz high-pass filter is applied to the EEG recordings. The Fully Automated Statistical Thresholding for EEG artifact Rejection (FASTER) algorithm rejects the bad channels and trials with greater than +/- 3 Z-scores on key metrics and pop_rejchan function from EEGLAB. Bad channels are interpolated except the mid-frontal Cz channel, which is never interpolated. Eye blink artifacts are removed following independent component analysis(ICA). We perform secondary preprocessing to match the pipeline of our method.

For the trials in each subject, we first align the channels into 19 channels Fp1, Fp2, F7, F3, Fz, F4, F8, T3, C3, Cz, C4, T4, T5, P3, Pz, P4, T6, O1, and O2 by selecting from the existing channels in the data. Channel Pz is not included in the existing channels, so we use the closest channel, POz, as a replacement. Besides, T7, T8, P7, P8 are the same channels as T3, T4, T5, T6. Finally, we select Fp1, Fp2, F7, F3, Fz, F4, F8, T7, C3, Cz, C4, T8, P7, P3, POz, P4, P8, O1, and O2 as the 19 standard channels. After alignment, we obtain recording trials with all 19 standard channels. We then downsample all trials from 500Hz to 128Hz. Next, we segment the trials into 1-second, non-overlapping samples, excluding those shorter than 1 second at the edges of the trial. This results in a total of 23,839 1-second, 128Hz samples. We apply bandpass filtering in the range of 0.5Hz to 45Hz, followed by standard normalization on each channel. This 19-channel processed dataset is used for self-supervised contrastive pretraining. Additionally, we process a raw 60-channel data version for supervised learning on individual datasets by not performing channel alignment while keeping all other procedures the same. Since some trials have mismatched channels, the 60 channels are picked by the most common channel names among trials.

### D.2.5. REEG-SRM.

The REEG-SRM (SRM Resting-state EEG) dataset is a publicly available EEG dataset on the OpenNEURO website[19], referenced in the paper (Hatlestad-Hall et al., 2022). This dataset contains resting-state EEG extracted from the experimental paradigm used in the Stimulus-Selective Response Modulation (SRM) project at the Department of Psychology, University of Oslo, Norway. The EEG data are recorded using 64 electrodes with a BioSemi ActiveTwo system, following the positional scheme of the 10-10 system. The dataset includes 111 healthy control subjects, with some subjects having one trial and others having multiple trials. The sampling rate is set to 1024Hz. Preprocessing steps are applied to the raw data, including bad channel interpolation, artifact rejection, and bandpass filtering from 1Hz to 45Hz. We perform secondary preprocessing to match the pipeline of our method.

We exclude two subjects who cannot read, identified as "sub-029" and "sub-104." For the remaining 109 subjects, we perform channel alignment by selecting the 19 standard channels: Fp1, Fp2, F7, F3, Fz, F4, F8, T7, C3, Cz, C4, T8, P7, P3, Pz, P4, P8, O1, and O2. Note that T7, T8, P7, and P8 are the same channels as T3, T4, T5, and T6 in the international 10-20 and 10-10 systems. After alignment, we obtain recording trials with all 19 standard channels. We then downsample all trials from 1024Hz to 128Hz. Next, we segment the trials into 1-second, non-overlapping samples, excluding those shorter than 1 second at the edges of the trial. This results in a total of 32,760 1-second, 128Hz samples. We apply bandpass filtering in the range of 0.5Hz to 45Hz, followed by standard normalization on each channel. The 19-channel processed dataset is used for self-supervised contrastive pretraining. Additionally, we process a raw 64-channel data version for supervised learning on individual datasets by not performing channel alignment but keeping all other procedures the same. Since some trials have mismatched channels, the 60 channels are selected based on the most common channel names across trials.

### D.2.6. TDBRAIN.

The TDBrain (Two Decades-Brainclinics Research Archive for Insights in Neurophysiology) dataset[20][21], referenced in the paper (Van Dijk et al., 2022), is a large permission-available EEG time series dataset recording brain activities of 1274 subjects with 33 channels. Researchers need to send requests to the authors by filling out the application forms to get access to this dataset. This dataset aims to research neurological or psychiatric dysfunction, such as Major Depressive Disorder

---

[18]https://openneuro.org/datasets/ds004584/versions/1.0.0
[19]https://openneuro.org/datasets/ds003775/versions/1.2.1
[20]https://brainclinics.com/resources/
[21]https://www.synapse.org/Synapse:syn25671079/wiki/610278

(MDD), attention deficit hyperactivity disorder (ADHD), Subjective Memory Complaints (SMC), obsessive-compulsive disorder (OCD), Parkinson's disease (PD), and many other brain disorders. The EEG data is recorded in resting-states in eye-open and eye-closed states. The sampling rate is 500Hz. Preprocessing steps are applied to the raw data, including artifact rejection, 50Hz notch-frequency removal, and bandpass filtering from 0.5Hz to 100Hz. We perform secondary preprocessing to match the pipeline of our method.

We exclude subjects with "REPLICATION" and "NaN" indications in the datasets, which are left for validation and testing for the researcher's model by contacting them, as described in their paper. For the remaining 911 subjects, we perform channel alignment by selecting the 19 standard channels: Fp1, Fp2, F7, F3, Fz, F4, F8, T7, C3, Cz, C4, T8, P7, P3, Pz, P4, P8, O1, and O2. Note that T7, T8, P7, and P8 are the same channels as T3, T4, T5, and T6 in the international 10-20 and 10-10 systems. After alignment, we obtain recording trials with all 19 standard channels. We then downsample all trials from 500Hz to 128Hz. Next, we segment the trials into 1-second, non-overlapping samples, excluding those shorter than 1 second at the edges of the trial. This results in a total of 231,689 1-second, 128Hz samples. We apply bandpass filtering in the range of 0.5Hz to 45Hz, followed by standard normalization on each channel. The 19-channel processed dataset is used for self-supervised contrastive pretraining. Additionally, we process a raw 33-channel data version for supervised learning on individual datasets by not performing channel alignment but keeping all other procedures the same.

### D.2.7. TUEP.

The TUEP[22] is one of the datasets in The Temple University Hospital (TUH) Electroencephalography (EEG) Corpus, which is the world's largest open-source EEG corpus. Researchers can access this dataset by submitting a request via an application form to the authors. This dataset is a subset of TUEG and contains data from 100 subjects with epilepsy and 100 subjects without epilepsy, as determined by a certified neurologist. We perform data preprocessing to align the data with our method.

Each subject has one or more trials, and some trials may have different numbers of channels and sampling rates. We first select subjects who have 19 standard channels: Fp1, Fp2, F7, F3, Fz, F4, F8, T3, C3, Cz, C4, T4, T5, P3, Pz, P4, T6, O1, and O2, in all their trials. A total of 179 subjects meet this requirement. For each trial from these 179 subjects, the majority have a 256Hz sampling rate. We downsample or upsample all trials to 128Hz. Next, we segment the trials into 1-second, non-overlapping samples, excluding those shorter than 1 second at the trial edges. Some subjects have a large number of trials, resulting in more than 10,000 samples in total per subject. Since our goal for pretraining is to learn general EEG features and disease-related features across subjects, we aim to avoid the model overfitting to subject-specific features. Therefore, we set 800 as the maximum number of samples per subject, randomly selecting 800 samples if the total number exceeds this threshold. This results in a total of 143,200 1-second, 128Hz samples. We apply bandpass filtering in the range of 0.5Hz to 45Hz, followed by standard normalization on each channel.

## E. Implementation Details

*Table 3.* **Training Setups.** Training setups for our method and baselines, where the $\times$ indicates disabled and $\checkmark$ indicates enabled.

| Processed Datasets Version | Raw-Channel Datasets | Channel-Aligned Datasets | | |
|---|---|---|---|---|
| Setups / Methods | Single-Dataset Supervised | Unified Supervised | Pre-training | Unified Fine-tuning |
| TCN | $\checkmark$ | $\times$ | $\times$ | $\times$ |
| Transformer | $\checkmark$ | $\times$ | $\times$ | $\times$ |
| Conformer | $\checkmark$ | $\times$ | $\times$ | $\times$ |
| TimesNet | $\checkmark$ | $\times$ | $\times$ | $\times$ |
| Medformer | $\checkmark$ | $\times$ | $\times$ | $\times$ |
| LEAD-Vanilla(Ours) | $\checkmark$ | $\times$ | $\times$ | $\times$ |
| LEAD-Sup(Ours) | $\times$ | $\checkmark$ | $\times$ | $\times$ |
| LaBraM | $\times$ | $\times$ | $\times$ | $\checkmark$ |
| EEGPT | $\times$ | $\times$ | $\times$ | $\checkmark$ |
| TS2Vec | $\times$ | $\times$ | $\checkmark$ | $\checkmark$ |
| BIOT | $\times$ | $\times$ | $\checkmark$ | $\checkmark$ |
| EEG2Rep | $\times$ | $\times$ | $\checkmark$ | $\checkmark$ |
| LEAD-Base(Ours) | $\times$ | $\times$ | $\checkmark$ | $\checkmark$ |

---

[22]https://isip.piconepress.com/projects/nedc/html/tuh_eeg/

*Table 4.* **Training Parameters.** Detailed training parameters, including encoder layers, heads, model dimensions, and optimization setups. The – indicates a parameter not used. LaBraM and EEGPT are excluded from the table since the code structure is different.

| Methods / Params | backbone | e_layers | n_heads | d_model | d_ff | batch_size | train_epochs | optimizer | learning_rate | lr_scheduler | gradient_clip | patience | swa |
|---|---|---|---|---|---|---|---|---|---|---|---|---|---|
| **Single-Dataset Supervised Learning** | | | | | | | | | | | | | |
| TCN | TCN | 6 | – | – | – | 128 | 100 | AdamW | 1e-4 | CosineAnnealing | 4.0 | 15 | ✓ |
| Transformer | Transformer | 6 | 8 | 128 | 256 | 128 | 100 | AdamW | 1e-4 | CosineAnnealing | 4.0 | 15 | ✓ |
| Conformer | Conformer | 6 | 8 | 128 | 256 | 128 | 100 | AdamW | 1e-4 | CosineAnnealing | 4.0 | 15 | ✓ |
| TimesNet | TimesNet | 2 | – | 32 | 64 | 64 | 100 | AdamW | 1e-4 | CosineAnnealing | 4.0 | 15 | ✓ |
| Medformer | Medformer | 6 | 8 | 128 | 256 | 128 | 100 | AdamW | 1e-4 | CosineAnnealing | 4.0 | 15 | ✓ |
| LEAD-Vanilla(Ours) | LEAD | 12 | 8 | 128 | 256 | 128 | 100 | AdamW | 1e-4 | CosineAnnealing | 4.0 | 15 | ✓ |
| **Unified Supervised Learning** | | | | | | | | | | | | | |
| LEAD-Sup(Ours) | LEAD | 12 | 8 | 128 | 256 | 128 | 100 | AdamW | 1e-4 | CosineAnnealing | 4.0 | 15 | ✓ |
| **Self-Supervised Pre-training** | | | | | | | | | | | | | |
| TS2Vec | Transformer | 20 | 12 | 128 | 256 | 512 | 50 | AdamW | 2e-4 | CosineAnnealing | 4.0 | – | ✓ |
| BIOT | BIOT | 20 | 12 | 128 | 256 | 512 | 50 | AdamW | 2e-4 | CosineAnnealing | 4.0 | – | ✓ |
| EEG2Rep | EEG2Rep | 20 | 12 | 128 | 256 | 512 | 50 | AdamW | 2e-4 | CosineAnnealing | 4.0 | – | ✓ |
| LEAD-Base(Ours) | LEAD | 12 | 8 | 128 | 256 | 512 | 50 | AdamW | 2e-4 | CosineAnnealing | 4.0 | – | ✓ |
| **Unified Fine-tuning** | | | | | | | | | | | | | |
| TS2Vec | Transformer | 20 | 12 | 128 | 256 | 128 | 100 | AdamW | 1e-4 | CosineAnnealing | 4.0 | 15 | ✓ |
| BIOT | BIOT | 20 | 12 | 128 | 256 | 128 | 100 | AdamW | 1e-4 | CosineAnnealing | 4.0 | 15 | ✓ |
| EEG2Rep | EEG2Rep | 20 | 12 | 128 | 256 | 512 | 50 | AdamW | 2e-4 | CosineAnnealing | 4.0 | – | ✓ |
| LEAD-Base(Ours) | LEAD | 12 | 8 | 128 | 256 | 128 | 100 | AdamW | 1e-4 | CosineAnnealing | 4.0 | 15 | ✓ |

## E.1. LEAD

We perform pretraining on 11 datasets: **AD-Auditory** (Lahijanian et al., 2024), **ADFSU** (Vicchietti et al., 2023), **ADSZ** (Alves et al., 2022; Pineda et al., 2020), **APAVA** (Escudero et al., 2006; Smith et al., 2017), **Depression** (Cavanagh et al., 2019; Cavanagh, 2021), **PEARL-Neuro** (Dzianok & Kublik, 2024), **REEG-BACA** (Getzmann et al., 2024), **REEG-PD** (Singh et al., 2023), **REEG-SRM** (Hatlestad-Hall et al., 2022), **TDBrain** (Van Dijk et al., 2022), and **TUEP** (Veloso et al., 2017), and fine-tuning on 5 downstream datasets: **ADFTD** (Miltiadous et al., 2023b;a), **Brain-Lat** (Prado et al., 2023), **CNBPM** (Ieracitano et al., 2019b; Amezquita-Sanchez et al., 2019), **Cognision-ERP** (Cecchi et al., 2015), and **Cognision-rsEEG**. The details of datasets and preprocessing steps are described in the previous section D. The pretraining datasets include 7 non-AD brain diseases or healthy subjects and 4 AD datasets, totaling 2,354 subjects and 1,165,361 1-second, 128Hz samples. All the downstream datasets are binary classifications between Alzheimer's disease and healthy controls, totaling 615 subjects and 223,039 1-second, 128Hz samples. The rationale for selecting these datasets for pretraining or fine-tuning is presented in 2.2.

We compare our method with 10 baselines, including 5 supervised, 3 self-supervised learning, and 2 large EEG foundational models. These selected baselines are state-of-the-art methods or have shown strong performance in EEG or time series classification tasks. The 5 supervised learning methods include **TCN** (Bai et al., 2018), vanilla **Transformer** (Vaswani et al., 2017), **Conformer** (Song et al., 2022), **TimesNet** (Wu et al., 2023), and **Medformer** (Wang et al., 2024c). The 3 self-supervised learning methods are **TS2Vec** (Yue et al., 2022), **BIOT** (Yang et al., 2024), and **EEG2Rep** (Mohammadi Foumani et al., 2024). The 2 large EEG foundational models are **LaBraM** (Jiang et al., 2024) and **EEGPT** (Wang et al., 2024a).

Our method has three variants based on training setups: **LEAD-Vanilla(3.21M)**, **LEAD-Sup(3.21M)**, and **LEAD-Base(3.41M)**. The LEAD-Vanilla model is trained fully supervised on a single dataset without channel alignment, such as the 7-channel version of the Cognision-ERP dataset. LEAD-Sup and LEAD-Base use datasets with alignment to 19 channels. LEAD-Sup is the model trained unified supervised on 5 AD datasets together without pre-training. For LEAD-Base, we first perform self-supervised pre-training on 11 pre-training datasets. The trained model is then used for unified fine-tuning on 5 downstream AD datasets. Note that for both LEAD-Sup and LEAD-Base, the 5 downstream AD datasets are unified trained and evaluated in one run, which is different from the usual approach where supervised training or fine-tuning occurs on a single dataset. The five supervised learning baselines, including TCN, Transformer, Conformer, TimesNet, and Medformer, use the same setup as LEAD-Vanilla. The three self-supervised learning baselines, including TS2Vec, BIOT, and EEG2Rep, follow LEAD-Base's setup. For the two large EEG foundational models, LaBraM and EEGPT, we load their pre-trained models and use the same unified fine-tuning setup as our LEAD-Base. The training setups, including single-dataset or unified training, whether the training is fully supervised or self-supervised, and whether the channel-aligned dataset version is used are summarized in Table 3.

All baseline methods and our method's variants, except for LaBraM and EEGPT, are trained using the same code framework and pipelines. The training epoch for self-supervised pretraining is fixed at 50 epochs, with no early stopping mechanism.

Linear probing is applied on all downstream datasets every five epochs to monitor relative performance. For fully supervised learning or fine-tuning, the training epoch is set to 100, with early stopping after 15 epochs of patience based on the best F1 score. The batch sizes for pretraining, fully supervised learning, and fine-tuning are set to 512, 128, and 128, respectively. The optimizer used is AdamW. The initial learning rates for pretraining, fully supervised learning, and fine-tuning are set to 0.0002, 0.0001, and 0.0001, respectively, with the CosineAnnealingLR learning scheduler. Gradient norm clipping is set to 4.0, and Stochastic Weight Averaging (SWA)(Izmailov et al., 2018) is enabled to improve cross-subject representation learning. The parameters are summarized in Table 4.

For LaBraM and EEGPT, we use their public code frameworks and load their pre-trained models for fine-tuning, as both are large EEG foundational models. The selection of pre-training datasets is also an integral part of their method. The five fine-tuning AD datasets are all preprocessed to the same shape and sampling frequency to align with their pre-trained models as described in their paper. Further training details for these two methods are provided in the following subsections for each respective method.

We employ four evaluation metrics: sample-level accuracy, sample-level F1 score (macro-averaged), subject-level accuracy, and subject-level F1 score (macro-averaged) after majority voting, as described in 2.6. In the self-supervised pre-training stage, all subjects in the pre-training datasets are used for training. In the supervised learning or fine-tuning classification stage, the training, validation, and test sets are split based on the subject-independent setup with a ratio of 6:2:2 for each dataset, where each subject appears exclusively in one of these three sets. There is no dataset overlapping between the pretraining and finetuning datasets. The training process is conducted with five random seeds (41-45) on fixed training, validation, and test sets to compute the mean and standard deviation of the models. All experiments are run on an NVIDIA RTX 4090 GPU and a server with 4 RTX A5000 GPUs, using Python 3.8 and PyTorch 2.0.0 + cu118.

## E.2. TCN

Temporal Convolutional Networks (TCN) (Bai et al., 2018) are a specialized type of convolutional network designed for time series tasks such as forecasting and classification. TCNs use causal dilated convolutions to expand the receptive field of the network while preventing information leakage from the past. Based on our experience, TCNs typically offer fast training speeds and relatively good performance in many time series classification tasks (Yue et al., 2022; Wang et al., 2024b), including EEG classification. This is a fully supervised method, and we train it on datasets without channel alignment. We set $e\_layers$ = 6. The method specified parameters are $hidden\_dims$ = 128, $output\_dims$ = 320, and $kernel\_size$ = 3.

## E.3. Transformer

Transformer (Vaswani et al., 2017), commonly known as the vanilla transformer, is introduced in the well-known paper "Attention is All You Need." It can also be applied to time series by embedding each cross-channel timestamp as a token and performing self-attention among these input tokens. This is a fully supervised method, and we train it on datasets without channel alignment. We set $e\_layers$ = 6, $n\_heads$ = 8, $d\_model$ = 128, and $d\_ff$ = 256.

## E.4. Conformer

EEG-Conformer (Song et al., 2022) is specifically designed for EEG classification by combining convolutional networks and self-attention modules. They first use convolutional modules to learn low-level local features and embeds the raw data into patches. A self-attention module is applied to these patches to capture global features. This is a fully supervised method, and we train it on datasets without channel alignment. We set $e\_layers$ = 6, $n\_heads$ = 8, $d\_model$ = 128, and $d\_ff$ = 256.

## E.5. TimesNet

TimesNet (Wu et al., 2023) is designed for general time series analysis. Instead of using 1D raw time series data, it first transforms the data into a 2D format based on multiple periods. This transformation embeds intra-period and inter-period variations into the columns and rows of the 2D tensors, respectively, to capture more robust features with 2D convolutions. According to a recent survey (Wang et al., 2024f), TimesNet achieves the best classification performance in many time series benchmarks. This is a fully supervised method, and we train it on datasets without channel alignment. We set $e\_layers$ = 2, $d\_model$ = 32, and $d\_ff$ = 64. The method specified parameter $top\_k$ is set to 3.

### E.6. Medformer

Medformer (Wang et al., 2024c) is designed for biomedical time series classification, including EEG and ECG. They utilize three mechanisms: cross-channel patching, multi-granularity embedding, and intra-inter granularity self-attention. These mechanisms enable it to capture channel correlations and multi-granularity temporal features effectively. This is a fully supervised method, and we train it on datasets without channel alignment. We set $e\_layers$ = 6, $d\_model$ = 128, and $d\_ff$ = 256. The method specified parameters $patch\_len\_list$ is set to [2, 4, 8].

### E.7. TS2Vec

TS2Vec (Yue et al., 2022) is a well-known self-supervised contrastive method designed for time series analysis. They effectively capture fine-grained features in time series at the timestamp level. Unlike other contrastive frameworks in computer vision domains (e.g., SimCLR, MOCO), which compute contrastive loss on denser representations after the projection head, TS2Vec computes the contrastive loss directly on the representations after the encoder, at both the timestamp and sample levels. Since the loss computation for each timestamp is required in this method, backbone models that fuse timestamps into patches are not suitable. Therefore, we replace the TCN backbone model used in their paper with a vanilla Transformer to better align with the large foundation model training. We pre-train on 11 datasets and then perform unified fine-tuning on the five downstream AD datasets together. We set $e\_layers$ = 20, $n\_heads$ = 12, $d\_model$ = 128, and $d\_ff$ = 256.

### E.8. BIOT

BIOT (Yang et al., 2024) is the first large foundation model for biosignals. They employ single-channel patching techniques to handle biosignals with varying numbers of channels. Each patch is mapped into tokens, with segment embedding, channel embedding, and positional embedding added to incorporate channel and positional information, making the tokens distinguishable from each other. A self-supervised contrastive framework is used to pretrain the model. We first pretrain on 11 datasets and then perform unified fine-tuning on the five downstream AD datasets. We set $e\_layers$ = 20, $n\_heads$ = 12, $d\_model$ = 128, and $d\_ff$ = 256.

### E.9. EEG2Rep

EEG2Rep (Mohammadi Foumani et al., 2024) is a self-supervised learning method that uses context-level masking and reconstruction instead of raw data-level reconstruction. The raw data is embedded into patches, and the masking operations are performed on the patch embeddings. They employ two networks: the context network, which is used as a query, and the target network, which serves as a key for calculating the L2 loss. A cross-attention predictor is used to align the output shapes of the context and target networks. We first pre-train on 11 datasets and then perform unified fine-tuning on the five downstream AD datasets. We set $e\_layers$ = 20, $n\_heads$ = 12, $d\_model$ = 128, and $d\_ff$ = 256.

### E.10. LaBraM

LaBraM (Jiang et al., 2024) is the first large foundational model for EEG. They design three-step pre-training strategies. They first pre-train a vector-quantified neural code book that encodes single-channel EEG patches into compact neural codes representations. Then, they pre-train a neural transformer by predicting the original neural codes for the masked EEG patches. Last, the encoder part of the pre-trained neural transformer is reused and a new classification head is added for finetuning on new datasets. We use the base version of the model checkpoint with $e\_layers$ = 12, $n\_heads$ = 10, $d\_model$ = 200, and $d\_ff$ = 800. We preprocess the five downstream AD datasets into 8-second, 1600 timestamps, and 200Hz samples to match the pre-trained model of their methods.

### E.11. EEGPT

EEGPT (Wang et al., 2024a) is a state-of-the-art large foundational model for EEG. They design a combination of an alignment loss between encoded tokens and momentum-encoded tokens and a reconstruction loss between reconstructed patches and masked patches. A spatiotemporal embedding is used to encode single-channel patches and by adding channel embedding and patch embedding together. The pre-trained model we used is the large version, which has $e\_layers$ = 8, $n\_heads$ = 8, $d\_model$ = 512, and $d\_ff$ = 2048. We preprocess the five downstream AD datasets into 4-second, 1024 timestamps, and 200Hz samples to match the pre-trained model of their methods.

# F. Ablation Study

## F.1. Ablation Study of Non-AD Datasets

*Table 5.* **Ablation Study of Non-AD Datasets.** This table presents the result change by adding more non-AD datasets during pre-training.

| Datasets | ADFTD (53,215 Samples) (65 Subjects) | | BrainLat (29,788 Samples) (67 Subjects) | | CNBPM (46,336 Samples) (126 Subjects) | | Cognision-ERP (61,300 Samples) (177 Subjects) | | Cognision-rsEEG (32,400 Samples) (180 Subjects) | |
|---|---|---|---|---|---|---|---|---|---|---|
| **Sample-Level Classification** | | | | | | | | | | |
| Metrics / #-Datasets | Accuracy | F1 Score | Accuracy | F1 Score | Accuracy | F1 Score | Accuracy | F1 Score | Accuracy | F1 Score |
| 5 | **78.07±1.16** | **77.99±1.14** | 74.91±0.47 | 74.85±0.44 | 94.25±0.81 | 92.55±1.08 | 67.28±0.72 | 67.20±0.72 | 73.67±0.69 | 73.38±0.75 |
| 7 | 76.82±0.62 | 76.77±0.60 | 74.89±1.49 | 74.81±1.52 | 95.40±0.42 | 94.12±0.54 | 67.75±0.76 | 67.69±0.75 | 74.52±0.24 | 74.28±0.22 |
| 9 | 76.84±0.61 | 76.81±0.60 | 74.70±1.13 | 74.63±1.17 | 96.09±0.30 | 94.99±0.40 | 67.54±0.75 | 67.46±0.73 | 75.32±0.28 | 75.08±0.28 |
| 11 | 76.64±0.87 | 76.64±0.86 | **77.89±1.28** | **77.80±1.34** | **96.51±0.33** | **95.53±0.42** | **69.58±0.90** | **69.53±0.91** | **76.21±0.39** | **76.01±0.39** |
| **Subject-Level Classification** | | | | | | | | | | |
| Metrics / #-Datasets | Accuracy | F1 Score | Accuracy | F1 Score | Accuracy | F1 Score | Accuracy | F1 Score | Accuracy | F1 Score |
| 5 | **84.29±2.86** | **84.26±2.90** | 84.29±2.86 | 84.26±2.90 | 93.85±3.08 | 93.84±3.08 | 73.33±2.22 | 73.29±2.20 | 83.78±0.00 | 83.52±0.08 |
| 7 | 82.86±3.50 | 82.81±3.55 | 84.29±2.86 | 84.26±2.90 | 96.92±1.54 | 96.92±1.54 | 73.33±2.83 | 73.32±2.84 | 86.49±1.71 | 86.32±1.78 |
| 9 | 81.43±3.50 | 81.36±3.55 | 84.29±2.86 | 84.26±2.90 | 99.23±1.54 | 99.23±1.54 | 71.67±2.08 | 71.62±2.03 | 85.41±1.32 | 85.23±1.35 |
| 11 | 80.00±5.35 | 79.96±5.36 | **90.00±3.50** | **89.98±3.48** | **100.00±0.00** | **100.00±0.00** | **84.44±2.22** | **84.42±2.21** | **91.89±1.71** | **91.86±1.73** |

*Figure 4.* **Ablation Study of Non-AD datasets.** It shows the performance change when adding more non-AD datasets in pre-training.

**Setup.** We conduct an ablation study to evaluate the performance changes when adding more Non-AD datasets to self-supervised contrastive pretraining. We begin by pretraining on five datasets: ADSZ, APAVA, ADFSU, AD-Auditory, and TDBRAIN, and fine-tuning on five datasets: ADFTD, CNBPM, Cognision-rsEEG, Cognision-ERP, and BrainLat, following the same setups as the model LEAD-Base. The **P** denotes pretraining, and **F** denotes fine-tuning, with the following number indicating the number of datasets used. We then gradually add two more pretraining datasets in the following pairs for the ablation study: (TUEP, REEG-PD), (PEARL-Neuro, Depression), and (REEG-SRM, REEG-BACA). The model uses 11 pre-training datasets is the same as LEAD-Base in Table 2. This study aims to investigate whether adding more Non-AD datasets to pre-training improves the model's ability to discriminate AD patterns during downstream fine-tuning.

**Results.** The results are presented in Table 5, and a figure showing the performance changes is displayed in Figure 4. We observe that the overall performance at both the subject- and sample-level improves for four out of the five datasets, excluding ADFTD. Performance increases for the last two pretraining datasets, with dramatic improvements seen in BrainLat, Cognision-ERP, and Cognision-rsEEG. Specifically, the subject-level F1 scores improved by 5.72%, 12.8%, and 6.63%, respectively. This improvement is intuitive, as the last two added datasets, REEG-SRM and REEG-BACA, have the largest number of samples, totaling more than 600K samples. The performance improvement in Cognision-ERP and Cognision-rsEEG also demonstrates that self-supervised pretraining benefits datasets, especially those with fewer channels (7 in the raw datasets), helping them distinguish general EEG or other brain disease features from AD-specific features.

For datasets that initially demonstrated good performance, such as CNBPM, adding more Non-AD datasets to pretraining results in gradual improvements, with a total improvement of 6.16% in the F1 score from 5 to 11 pretraining datasets. As for the performance drop in ADFTD, it is consistent with the finding in Table 2 that the performance of LEAD-Sup using supervised learning is much better than that of LEAD-Base using self-supervised learning. The reasons for this drop are unclear and require further investigation in future research.

*Table 6.* **Ablation Study of AD Datasets.** This table presents the results of ADFTD by adding more AD datasets in supervised learning.

| Datasets | ADFTD (53,215 Samples, 65 Subjects) | | | |
|---|---|---|---|---|
| | Sample-Level Classification | | Subject-Level Classification | |
| #-Datasets \ Metrics | Accuracy | F1 Score | Accuracy | F1 Score |
| S-1-V-ADFTD-Sup | 73.81±1.02 | 73.75±1.00 | 82.86±3.50 | 82.81±3.55 |
| S-2-V-ADFTD-Sup | 77.91±1.52 | 77.73±1.50 | 77.14±2.86 | 76.94±3.05 |
| S-3-V-ADFTD-Sup | 81.31±1.04 | 80.84±0.93 | 87.14±2.86 | 86.25±3.15 |
| S-4-V-ADFTD-Sup | 82.18±1.06 | 81.76±0.89 | **90.00±3.50** | **89.35±4.00** |
| S-5-V-ADFTD-Sup | **82.79±0.71** | **82.39±0.73** | 87.14±2.86 | 86.31±3.27 |

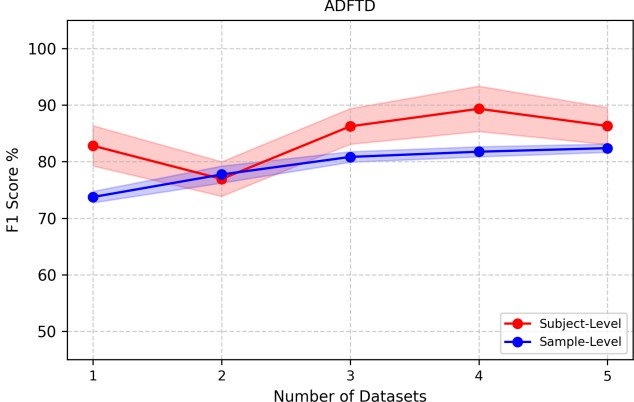

*Figure 5.* **Ablation Study of AD datasets.** It shows the results change of ADFTD when adding more AD datasets in supervised learning.

### F.2. Ablation Study of AD Datasets

**Setup.** We conduct an ablation study to evaluate the performance changes when adding more AD datasets into unified supervised learning. All AD datasets are added to the training set one by one in the following order: ADFTD, BrainLat, CNBPM, Cognision-ERP, and Cognision-rsEEG, with validation and testing conducted on a single dataset: ADFTD. The models are named from S-1-V-ADFTD-Sup to S-5-V-ADFTD-Sup, where **S** denotes supervised learning and **V** denotes validation, with the following number indicating the number of datasets used. Note that the performance of S-5-V-ADFTD-Sup on ADFTD may differ slightly from LEAD-Sup in Table 2, as S-5-V-ADFTD-Sup validates only on ADFTD to select the best-performing model, rather than selecting the model with the best weighted F1 score across all five datasets. This study investigates whether high-quality AD datasets can benefit each other in unified supervised learning compared to single-dataset supervised learning, even without self-supervised pretraining.

**Results.** The results are presented in Table 6, and a figure showing the performance changes is displayed in Figure 5. We observe a gradual improvement in performance at the sample-level classification, with the F1 score increasing from 73.75 to 82.39. For subject-level results, the overall trend is upward. However, we believe the fluctuations are due to the limited number of subjects compared to the total number of samples. Additionally, the imbalanced number of samples per subject in the ADFTD dataset may contribute to the larger variability observed.

### F.3. Contrastive Learning Modules

**Setup.** We conduct an ablation study to investigate the functionality of each contrastive module. Specifically, we use either sample-level contrast or subject-level contrast for self-supervised pre-training, while keeping the other setups the same as the LEAD-Base model. Recall that sample-level contrast involves instance discrimination, where different views of the same sample are treated as positive pairs, and the rest are treated as negative pairs, similar to setups used in other domains like computer vision. In contrast, subject-level contrast is specifically designed for biomedical time series data, such as EEG, which contains subject information. We consider samples segmented from the same subject as positive pairs and samples from different subjects as negative pairs, thus performing a subject-discrimination task. We believe subject-level contrast helps to reduce subject-specific noise, leading to more robust general feature learning. This approach makes the embedding space more uniform across subjects and improves inter-subject classification in downstream tasks.

*Table 7.* **Ablation Study of Contrastive Learning Modules.** This table presents the ablation study of each module in self-supervised contrastive learning during the pretraining stage. The **All** is the same as the LEAD-Base model.

| Datasets | ADFTD (53,215 Samples) (65 Subjects) | | BrainLat (29,788 Samples) (67 Subjects) | | CNBPM (46,336 Samples) (126 Subjects) | | Cognision-ERP (61,300 Samples) (177 Subjects) | | Cognision-rsEEG (32,400 Samples) (180 Subjects) | |
|---|---|---|---|---|---|---|---|---|---|---|
| **Sample-Level Classification** | | | | | | | | | | |
| Modules / Metrics | Accuracy | F1 Score | Accuracy | F1 Score | Accuracy | F1 Score | Accuracy | F1 Score | Accuracy | F1 Score |
| Sample-Level Contrast | **78.67**±**1.47** | **78.55**±**1.43** | 70.34±0.43 | 70.27±0.40 | 93.75±0.29 | 92.04±0.38 | 64.87±0.62 | 64.67±0.66 | 70.28±0.67 | 69.65±0.84 |
| Subject-Level Contrast | 75.60±2.43 | 75.55±2.48 | 75.73±1.53 | 75.65±1.53 | **96.51**±**0.21** | **95.55**±**0.28** | **69.78**±**0.46** | **69.69**±**0.46** | 75.74±0.31 | 75.53±0.30 |
| All | 76.64±0.87 | 76.64±0.86 | **77.89**±**1.28** | **77.80**±**1.34** | 96.51±0.33 | 95.53±0.42 | 69.58±0.90 | 69.53±0.91 | **76.21**±**0.39** | **76.01**±**0.39** |
| **Subject-Level Classification** | | | | | | | | | | |
| Models / Metrics | Accuracy | F1 Score | Accuracy | F1 Score | Accuracy | F1 Score | Accuracy | F1 Score | Accuracy | F1 Score |
| Sample-Level Contrast | **81.43**±**3.50** | **81.36**±**3.55** | 82.86±3.50 | 82.81±3.55 | 96.15±0.00 | 96.15±0.00 | 78.33±2.08 | 78.31±2.08 | 80.54±2.02 | 80.03±2.13 |
| Subject-Level Contrast | 81.43±3.50 | 81.36±3.55 | 87.14±5.35 | 87.11±5.36 | **100.00**±**0.00** | **100.00**±**0.00** | 82.22±3.33 | 82.21±3.33 | 90.27±1.32 | 90.23±1.34 |
| All | 80.00±5.35 | 79.96±5.36 | **90.00**±**3.50** | **89.98**±**3.48** | 100.00±0.00 | 100.00±0.00 | **84.44**±**2.22** | **84.42**±**2.21** | **91.89**±**1.71** | **91.86**±**1.73** |

**Results.** The results are presented in Table 7. The **All** is the same as the LEAD-Base model. We observe that subject-level contrast significantly outperforms sample-level contrast for four out of five datasets (except for ADFTD), with around a 5% improvement in F1 score for subject-level classification. Subject-level contrast performs poorly on ADFTD, likely due to unknown factors, and this could be the reason that adding self-supervised pre-training does not work well for ADFTD. We plan to investigate this in future work and explore alternative self-supervised training methods for the ADFTD dataset, such as mask and reconstruction. Overall, in most cases, subject-level contrast performs better than sample-level contrast, and using both contrastive modules together achieves comparable or better results than subject-level contrast alone.

## F.4. Finetune or Validate on One Dataset

*Table 8.* **Single-Dataset Finetuning or Validation.** This table presents the results of single-dataset fine-tuning or unified fine-tuning but validated on the single dataset. Two models are named with P-11-F-1-Base and P-11-F-5-V-1-Base.

| Datasets | ADFTD (53,215 Samples) (65 Subjects) | | BrainLat (29,788 Samples) (67 Subjects) | | CNBPM (46,336 Samples) (126 Subjects) | | Cognision-ERP (61,300 Samples) (177 Subjects) | | Cognision-rsEEG (32,400 Samples) (180 Subjects) | |
|---|---|---|---|---|---|---|---|---|---|---|
| **Sample-Level Classification** | | | | | | | | | | |
| Models / Metrics | Accuracy | F1 Score | Accuracy | F1 Score | Accuracy | F1 Score | Accuracy | F1 Score | Accuracy | F1 Score |
| P-11-F-1-Base(3.41M) | **77.30**±**0.93** | **77.28**±**0.93** | **78.97**±**0.53** | **78.92**±**0.56** | **96.78**±**0.34** | **95.93**±**0.44** | 60.72±0.61 | 60.59±0.62 | 64.13±0.68 | 64.06±0.66 |
| P-11-F-5-V-1-Base(3.41M) | 76.55±0.63 | 76.54±0.63 | 78.45±0.85 | 78.39±0.89 | 96.68±0.34 | 95.74±0.45 | **70.13**±**0.55** | **70.08**±**0.56** | 75.95±0.26 | 75.76±0.28 |
| LEAD-Base(3.41M) | 76.64±0.87 | 76.64±0.86 | 77.89±1.28 | 77.80±1.34 | 96.51±0.33 | 95.53±0.42 | 69.58±0.90 | 69.53±0.91 | **76.21**±**0.39** | **76.01**±**0.39** |
| **Subject-Level Classification** | | | | | | | | | | |
| Models / Metrics | Accuracy | F1 Score | Accuracy | F1 Score | Accuracy | F1 Score | Accuracy | F1 Score | Accuracy | F1 Score |
| P-11-F-1-Base(3.41M) | 77.14±2.86 | 77.05±2.81 | 90.00±3.50 | 89.98±3.48 | 97.69±1.88 | 97.69±1.89 | 65.56±3.33 | 65.47±3.32 | 72.43±3.97 | 72.40±4.01 |
| P-11-F-5-V-1-Base(3.41M) | 78.57±4.52 | 78.51±4.52 | 87.14±2.86 | 87.14±2.84 | 100.00±0.00 | 100.00±0.00 | **85.00**±**2.22** | **84.97**±**2.22** | 91.89±1.71 | 91.86±1.73 |
| LEAD-Base(3.41M) | 80.00±5.35 | 79.96±5.36 | 90.00±3.50 | 89.98±3.48 | 100.00±0.00 | 100.00±0.00 | 84.44±2.22 | 84.42±2.21 | 91.89±1.71 | 91.86±1.73 |

**Setup.** We conduct experiments that fine-tune on a single dataset and fine-tune on multiple datasets while validating on a single dataset. In the LEAD-Base model, we perform unified fine-tuning, which involves fine-tuning five downstream AD datasets simultaneously and using the mean F1 score across all five datasets for early stopping based on the best performance. We aim to compare the functionality of unified fine-tuning against single-dataset fine-tuning, and investigate whether validating on a specific dataset could improve performance. Two new models are named with P-11-F-1-Base and P-11-F-5-V-1-Base. Other setups are the same as the LEAD-Base model.

**Results.** The results are presented in Table 8. For ADFTD, BrainLat, and CNBPM, we observe a performance improvement in sample-level classification when using single-dataset fine-tuning. However, fine-tuning across the five AD datasets together still yields better performance in subject-level classification on these three datasets. This supports the idea that unified fine-tuning leads to more balanced results among subjects, reducing the likelihood of overfitting to specific subjects. For the two Cognision datasets, performance drops significantly when fine-tuning on a single dataset. Since the raw channels

for these two datasets are fewer compared to other datasets and inherently contain less information, we can infer that datasets with fewer channels rely more heavily on unified fine-tuning.

When validating on a single dataset, we observe no improvement, except for a subtle improvement in Cognision-ERP. We believe this is due to using the average metrics across datasets for early stopping, which helps alleviate the risk of overfitting to the validation set on specific datasets, leading to more robust performance on the test set.

## F.5. Public Datasets Training

*Table 9.* **Public Datasets Training.** This table presents the training results only on the public datasets. We follow the same training strategy as LEAD-Sup and LEAD-Base but use only two public AD datasets. The new models are named with S-2-Sup and P-11-F-2-Base.

| Datasets | ADFTD (53,215 Samples) (65 Subjects) | | BrainLat (29,788 Samples) (67 Subjects) | |
|---|---|---|---|---|
| | **Sample-Level Classification** | | | |
| Metrics / Models | **Accuracy** | **F1 Score** | **Accuracy** | **F1 Score** |
| **S-2-Sup(3.21M)** | 77.80±1.40 | 77.67±1.42 | 71.91±0.56 | 71.91±0.56 |
| **P-11-F-2-Base(3.41M)** | 77.78±1.20 | 77.77±1.20 | 77.66±1.19 | 77.59±1.20 |
| **LEAD-Vanilla(3.21M)** | 73.81±1.02 | 73.75±1.00 | 62.15±1.28 | 62.07±1.28 |
| **LEAD-Sup(3.21M)** | **80.84±0.84** | **80.68±0.79** | 70.36±0.62 | 70.31±0.65 |
| **LEAD-Base(3.41M)** | 76.64±0.87 | 76.64±0.86 | **77.89±1.28** | **77.80±1.34** |
| | **Subject-Level Classification** | | | |
| Metrics / Models | **Accuracy** | **F1 Score** | **Accuracy** | **F1 Score** |
| **S-2-Sup(3.21M)** | 82.86±5.71 | 82.74±5.95 | 80.00±2.86 | 79.91±2.90 |
| **P-11-F-2-Base(3.41M)** | 82.86±3.50 | 82.81±3.55 | 88.57±3.50 | 88.56±3.48 |
| **LEAD-Vanilla(3.21M)** | 82.86±3.50 | 82.81±3.55 | 75.71±5.71 | 75.39±5.78 |
| **LEAD-Sup(3.21M)** | **91.43±2.86** | **91.34±2.81** | 78.57±0.00 | 78.46±0.00 |
| **LEAD-Base(3.41M)** | 80.00±5.35 | 79.96±5.36 | **90.00±3.50** | **89.98±3.48** |

**Setup.** We conduct an experiment where we train only on public datasets. Since all the pretraining datasets are public, we modify only the fine-tuning or unified fully supervised learning setups. Specifically, we use ADFTD and BrainLat for unified supervised learning by training the two datasets together, referred to as S-2-Sup. All other setups remain the same as the LEAD-Sup model. We also perform unified fine-tuning by fine-tuning the two datasets together, referred to as P-11-F-2-Base, with all other setups remaining the same as the LEAD-Base model.

**Results.** The results are presented in Table 9. We observe that unified supervised learning or fine-tuning still benefits performance compared to the LEAD-vanilla model, which is trained on a single dataset with raw channel numbers. The improvement is not as pronounced for ADFTD, but it is significant for BrainLat, with approximately 15% and 13% F1 score improvements in sample-level and subject-level classification, respectively.

# G. Supplementary Experiments

## G.1. Frequency Bands Analysis

**Setup.** We conduct experiments that fine-tune on AD datasets across different frequency bands. Recall that in the data preprocessing stage, we filter the frequency band from 0.5–45Hz. In this setup, the pretraining remains unchanged, but we filter the downstream AD datasets into different frequency bands. Other fine-tuning setups are the same as in the LEAD-Base model. Specifically, the bands are Delta ($\delta$) (0.5–4Hz), Theta ($\theta$) (4–7Hz), Alpha ($\alpha$) (8–12Hz), Beta ($\beta$) (12–30Hz), and Gamma ($\gamma$) (30–45Hz). We perform unified fine-tuning using the data filtered into these different bands.

**Results.** The results for different frequency bands are presented in Table 10. The **All** model is the same as the LEAD-Base model. We observe that the Theta ($\theta$) (4–7Hz), Alpha ($\alpha$) (8–12Hz), and Beta ($\beta$) (12–30Hz) bands are usually the most critical for classification performance. These three bands consistently yield the highest sample-level F1 scores, as well as strong performance in subject-level classification. However, there are two exceptions. First, for subject-level classification in BrainLat, the highest performance is achieved in the Gamma ($\gamma$) (30–45Hz) band. We speculate that this may be due to the imbalanced number of subjects per sample, which could cause overfitting to specific subjects. The second exception is the

*Table 10.* **Frequency Bands Analysis.** This table presents the results of fine-tuning on different frequency bands. We use **bold** and underline to highlight the most and second important bands, respectively. The results for all bands are presented as a reference.

| Datasets | ADFTD (53,215 Samples) (65 Subjects) | | BrainLat (29,788 Samples) (67 Subjects) | | CNBPM (46,336 Samples) (126 Subjects) | | Cognision-ERP (61,300 Samples) (177 Subjects) | | Cognision-rsEEG (32,400 Samples) (180 Subjects) | |
|---|---|---|---|---|---|---|---|---|---|---|
| **Sample-Level Classification** | | | | | | | | | | |
| Models \ Metrics | Accuracy | F1 Score | Accuracy | F1 Score | Accuracy | F1 Score | Accuracy | F1 Score | Accuracy | F1 Score |
| **Delta** $\delta$ **(0.5-4Hz)** | 65.97±0.63 | 65.69±0.66 | 59.78±0.82 | 59.65±0.77 | 85.21±0.41 | 80.23±0.58 | 59.06±0.40 | **58.68±0.35** | 60.66±0.49 | 59.95±0.48 |
| **Theta** $\theta$ **(4-7Hz)** | 68.09±0.76 | 67.97±0.69 | 61.52±0.41 | 61.23±0.36 | 88.73±0.17 | 85.07±0.25 | **59.13±0.44** | 58.59±0.56 | 65.54±0.68 | 64.67±0.66 |
| **Alpha** $\alpha$ **(8-12Hz)** | 66.60±0.77 | 66.57±0.76 | **67.32±0.54** | **67.17±0.53** | 93.13±0.27 | 91.15±0.35 | 57.94±0.40 | 57.57±0.35 | 63.59±0.26 | 62.94±0.20 |
| **Beta** $\beta$ **(12-30Hz)** | **69.65±0.39** | **69.29±0.41** | 65.51±1.19 | 65.47±1.16 | **93.92±0.46** | **92.14±0.64** | 54.20±0.78 | 53.90±0.79 | **67.11±0.24** | **66.67±0.33** |
| **Gamma** $\gamma$ **(30-45Hz)** | 62.32±0.81 | 62.14±0.88 | 64.69±6.55 | 64.65±6.55 | 84.16±0.48 | 78.08±0.93 | 51.17±2.01 | 50.90±1.94 | 57.38±0.48 | 57.03±0.62 |
| **All (0.5-45Hz)** | 76.64±0.87 | 76.64±0.86 | 77.89±1.28 | 77.80±1.34 | 96.51±0.33 | 95.53±0.42 | 69.58±0.90 | 69.53±0.91 | 76.21±0.39 | 76.01±0.39 |
| **Subject-Level Classification** | | | | | | | | | | |
| Models \ Metrics | Accuracy | F1 Score | Accuracy | F1 Score | Accuracy | F1 Score | Accuracy | F1 Score | Accuracy | F1 Score |
| **Delta** $\delta$ **(0.5-4Hz)** | 72.86±5.35 | 72.66±5.39 | 75.71±7.28 | 73.69±8.99 | 79.23±4.62 | 78.93±4.68 | **70.56±3.33** | **69.17±3.55** | 76.76±2.76 | 74.56±3.15 |
| **Theta** $\theta$ **(4-7Hz)** | 81.43±3.50 | 81.36±3.55 | 68.57±5.71 | 66.36±5.06 | 84.62±0.00 | 84.24±0.00 | 61.67±2.08 | 60.30±3.43 | 78.92±3.59 | 77.66±4.25 |
| **Alpha** $\alpha$ **(8-12Hz)** | **82.86±3.50** | **82.81±3.55** | 78.57±0.00 | 78.46±0.00 | 96.15±0.00 | 96.15±0.00 | 65.56±3.77 | 65.05±4.00 | **81.62±1.08** | **81.27±1.11** |
| **Beta** $\beta$ **(12-30Hz)** | 78.57±0.00 | 78.46±0.00 | 77.14±2.86 | 76.94±3.05 | **92.31±2.43** | **92.30±2.44** | 58.33±1.76 | 58.14±1.65 | 74.59±4.71 | 73.98±4.75 |
| **Gamma** $\gamma$ **(30-45Hz)** | 60.00±3.50 | 59.39±3.85 | **82.86±10.69** | **82.78±10.75** | 76.92±2.43 | 75.58±2.86 | 47.78±6.67 | 47.38±6.50 | 64.32±2.02 | 63.00±2.93 |
| **All (0.5-45Hz)** | 80.00±5.35 | 79.96±5.36 | 90.00±3.50 | 89.98±3.48 | 100.00±0.00 | 100.00±0.00 | 84.44±2.22 | 84.42±2.21 | 91.89±1.71 | 91.86±1.73 |

Cognision-ERP dataset, where the highest performance comes from the Delta ($\delta$) (0.5–4Hz) band. This result is plausible since Cognision-ERP is an event-related potentials dataset rather than resting-state EEG. Previous studies have shown that biomarkers in continuous attention tasks may manifest in this band (Kirmizi-Alsan et al., 2006).

## G.2. Channels Importance Analysis

*Table 11.* **Channels Importance Analysis.** This table presents the importance of different brain regions in the CNBPM dataset by masking specific channels during testing with a trained LEAD-Base model. We use **bold** and underline to highlight the most important and the second most important regions, respectively. Note that we define the most important regions as those that cause the greatest performance drop when masked. The results for all regions are presented as a reference.

| Datasets | CNBPM (46,336 Samples, 126 Subjects) | | | |
|---|---|---|---|---|
| | Sample-Level Classification | | Subject-Level Classification | |
| #-Datasets \ Metrics | Accuracy | F1 Score | Accuracy | F1 Score |
| **No Frontopolar** | 94.46±0.46 | 92.92±0.60 | 93.85±3.08 | 93.84±3.08 |
| **No Frontal** | **79.52±3.57** | **71.27±3.62** | **76.15±4.49** | **74.68±5.46** |
| **No Temporal** | 94.27±0.70 | 92.39±0.99 | 91.54±4.49 | 91.46±4.53 |
| **No Parietal** | 93.15±0.75 | 90.83±1.09 | 93.08±2.88 | 93.07±2.88 |
| **No Occipital** | 89.97±1.53 | 85.97±2.49 | 91.54±3.77 | 91.44±3.90 |
| **No Central** | 92.92±0.62 | 90.95±0.79 | 93.85±3.08 | 93.84±3.08 |
| **All Regions** | 96.51±0.33 | 95.53±0.42 | 100.00±0.00 | 100.00±0.00 |

**Setup.** We conduct experiments to assess the importance of different brain regions. Specifically, we keep the training stage the same, but mask the channels in specific regions during testing. The trained LEAD-Base model is used for this experiment. We perform this research on the CNBPM dataset, as it achieves the highest results and can mostly alleviate the interference of other factors, such as data quality. We define the masked region that causes the highest performance drop as the most critical region. The regions include: Frontopolar (Fp1, Fp2), Frontal (F7, F3, Fz, F4, F8), Temporal (T3, T4, T5, T6), Parietal (P3, Pz, P4), Occipital (O1, O2), and Central (C3, Cz, C4).

**Results.** The results are presented in Table 11. The **All Regions** is the same as the LEAD-Base model. We observe that the Frontal region is the most important, causing the most significant performance drop when masked, with a 24.26% and 25.32% F1 score reduction in sample-level and subject-level classification, respectively. The second most important region is the Occipital region, which causes a 9.56% and 8.56% F1 score drop in sample-level and subject-level classification. However, the performance drop here is not as pronounced as that seen with the Frontal region.

# H. Discussion

## H.1. Comparison With Existing Large EEG Model

To address the challenges posed by heterogeneous channels in EEG large foundational model pre-training, existing methods typically use single-channel patches as embedding tokens for transformers (Yang et al., 2024; Jiang et al., 2024; Wang et al., 2024a). However, this approach introduces two trade-offs concerning flexibility and computational resources.

First, there is a trade-off between model flexibility and unified dataset training. While single-channel patching allows the model to train on EEG data with varying numbers of channels, the token embedding methods are fixed, limiting the model to adopt backbone architectures different from BIOT (Yang et al., 2024). In such architectures, spatial features among channels can only be learned by attaching channel embeddings to each patch embedding. This approach restricts the model's ability to capture more prosperous spatial relationships, thus limiting the ability to extract spatial features.

Second, there is a trade-off between patch length and computational resources. With single-channel patch embedding, the channel number becomes a factor in determining the final number of input tokens. For instance, if each sample has 1024 timestamps and 19 channels, and the patch size is 32, the total number of patches would be $19 \times (1024 / 32) = 608$. As the patch size decreases, the number of patches increases, making the model computationally expensive to train. For example, if we choose a smaller patch length like 4, the number of patches increases significantly, which requires additional computational power to map all these patches into the d_model dimension tokens. This computational cost leads many existing methods to opt for larger patch sizes, such as 64 or 200 (Jiang et al., 2024; Wang et al., 2024a), which limits the model's ability to learn fine-grained temporal features.

To avoid these trade-offs, we perform channel alignment during data preprocessing. This strategy offers more flexibility for choosing the backbone model, reduces the computational burden, and enables unified fine-tuning on all downstream datasets in one run, which enhances model performance without compromising efficiency.

## H.2. Effectiveness of Subject-Level Contrast

We speculate that there are three potential reasons why subject-level contrastive pre-training significantly benefits downstream AD detection: **1)** Purification of Noise within the Subject: By treating all samples from a single subject as positive pairs, subject-level contrastive learning forces the model to make the sample representations within each subject more similar. This helps the model focus on the subject's inherent characteristics and purifies the noise caused by irrelevant subject-specific features such as artifacts. **2)** Learning General Features Associated with Subjects: Subject-level contrasting aims to differentiate subjects by pushing different subjects apart uniformly in the embedding space. This subject-discrimination task encourages the model to learn general features related to the subject, such as brain structure, age, gender, EEG devices, and brain health. These features are essential for downstream tasks where the goal is to classify AD based on subject-specific patterns. **3)** Compare with SimCLR: In computer vision, SimCLR is designed to perform sample-discrimination tasks for image classification during pre-training. In contrast, our ultimate goal is to classify subjects for AD detection. Treating each subject's EEG data as one "sample," subject-level contrastive learning becomes analogous to SimCLR, making it reasonable to perform subject-level contrasting during pre-training for better subject-level classification during downstream tasks.

## H.3. Limitations and Future Works

In this paper, to enable pre-training on various EEG datasets and perform unified fine-tuning, we align all the EEG data to 19 standard channels in data preprocessing. For datasets with more than 19 channels, we simply drop the extra channels, which may result in some potential information loss. However, we have demonstrated that this trade-off is manageable, as channel alignment still significantly benefits the overall training pipeline. Moving forward, we plan to explore methods to better utilize the information from additional channels, aiming to minimize any loss and enhance model performance. Additionally, while this study focuses on the potential of contrastive-based pre-training for AD detection, we also recognize the potential of other techniques, such as combining contrastive learning with mask-reconstruction modules or adopting a decoder-only architecture. These avenues will be explored in future works to enhance our model's performance.

