# OpenReview forum: "LEAD: Large Foundation Model for EEG-Based Alzheimer’s Disease Detection"
_ICML.cc/2025/Conference — Submitted to ICML 2025_

### Official Review · Reviewer_h4Kx · 2025-03-13

**Overall Recommendation:** 4

**Summary:**

This paper introduces LEAD, the first large foundation model for EEG-based Alzheimer’s disease detection, which overcomes challenges related to small dataset sizes and inter-subject variability by curating the largest EEG-AD corpus to date with 813 subjects from nine datasets. The proposed pipeline features robust data preprocessing, including channel and frequency alignment, segmentation, and normalization, followed by a novel self-supervised contrastive pre-training framework that employs both sample-level and subject-level contrastive learning to extract generalized EEG features. These features are subsequently fine-tuned using a unified multi-dataset approach with a majority voting scheme for subject-level classification. Experimental results reveal improvements of up to 9.86% in sample-level F1 score and 9.31% in subject-level F1 score over state-of-the-art methods, demonstrating the effectiveness of the subject-level pre-training and fine-tuning strategies in addressing inter-subject variations.

**Claims And Evidence:**

The submission’s claims are well-supported by clear and convincing evidence. The authors back their assertions with comprehensive experiments across multiple datasets and a robust comparison against state-of-the-art baselines, demonstrating significant improvements in both sample-level and subject-level F1 scores. Detailed ablation studies and analyses of the self-supervised pre-training modules, channel alignment, and unified fine-tuning further substantiate the effectiveness of their approach. Although some minor factors, such as potential dataset-specific variability, could be explored in more depth, the evidence provided convincingly supports the paper’s claims.

**Essential References Not Discussed:**

N.A.

**Experimental Designs Or Analyses:**

The experimental design is robust and thoughtfully constructed. The review examined the subject-independent evaluation setup, unified fine-tuning strategy across multiple EEG datasets, and the comprehensive comparisons against various sota baselines. The use of sample-level and subject-level metrics, along with detailed ablation studies, effectively isolates the contributions of components like channel alignment and contrastive pre-training. While additional statistical tests could further reinforce the findings, the overall experimental framework convincingly validates the proposed approach.

**Methods And Evaluation Criteria:**

The proposed methods and evaluation criteria are well-suited for the challenges of EEG-based Alzheimer’s detection. The paper introduces a comprehensive pipeline to mitigate inter-subject variability and data scarcity, including channel and frequency alignment, segmentation, normalization, and a novel self-supervised contrastive pre-training framework that operates at both sample and subject levels. Additionally, the evaluation metrics, which encompass both sample-level and subject-level F1 scores, along with extensive ablation studies and comparisons against state-of-the-art methods, provide a robust and practical means to assess performance. The design choices and benchmark datasets used are appropriate for the application at hand and effectively address the core issues of early Alzheimer’s detection using EEG.

**Other Comments Or Suggestions:**

N.A.

**Other Strengths And Weaknesses:**

Plz go and check the above comments.

**Questions For Authors:**

1. Could you provide more details on the dataset selection and curation process? In particular, how did you manage differences in data quality and subject demographics across the 9 datasets, and what measures were taken to mitigate potential biases?

2. In your self-supervised pre-training framework, how were the weighting coefficients (λ₁ and λ₂) for the sample-level and subject-level contrastive losses determined? Did you perform a sensitivity analysis on these hyperparameters, and if so, how did variations affect model performance?

3. Have you conducted any statistical significance tests to confirm that the performance improvements over the baseline methods are not due to chance?

4. Beyond the supplementary analyses on channel and frequency band importance, have you explored additional interpretability methods to link the model’s learned features with established EEG biomarkers of Alzheimer’s disease?

I will reconsider my assessment after checking authors' response.

**Relation To Broader Scientific Literature:**

The paper’s contributions are well-positioned within the broader scientific literature on EEG-based Alzheimer’s detection and self-supervised learning. Prior work in this domain largely focused on manual feature extraction, such as statistical, spectral, and complexity measures, or on applying typical deep learning methods, which were often hampered by small datasets and high inter-subject variability. In contrast, this paper leverages a foundation model approach by curating the largest EEG-AD dataset to date and employing self-supervised contrastive learning techniques at both the sample and subject levels. Additionally, the paper introduces unified fine-tuning with channel alignment and subject-independent evaluation, addressing known pitfalls such as data leakage common in subject-dependent setups.

**Theoretical Claims:**

The paper’s theoretical framework is well-grounded in established contrastive learning techniques. While it does not introduce entirely new formal proofs, the review confirms that these formulations are accurate and appropriate for addressing the challenges of EEG-based Alzheimer’s detection. By leveraging well-validated theoretical underpinnings, the authors effectively balance rigorous methodology with practical applicability, which is further supported by strong empirical results.

---

> ### Author Rebuttal · Authors · 2025-03-31
>
> Thank you for your thoughtful feedback on our work! We appreciate your careful reading and endorsement. Below, we address each of your questions and concerns:
>
> ---
>
> **Q1**: How manage dataset quality and potential demographic bias?
>
> **A1**:
> **Data Quality**
> For data quality, we initially surveyed existing EEG-AD studies, identifying both public and private datasets from our collaborators. We prioritized datasets that (1) had sufficient subject counts (≥50) and (2) exceeded 10k total 1-second samples for finetuning. Small datasets with uncertain quality (e.g., potential label shift, noisy data) were excluded in united finetuning (only used for pretraining). For the consistency of labels, the AD labels are officially diagnosed by physicians, thus are relatively robust with minimal label shift. If a patient is clinically diagnosed as AD, that label typically aligns well across different hospitals and cohorts, providing consistency across datasets.
>
> **Demographic Bias**
> Unfortunately, some datasets (APAVA, ADSZ, ADFSU) lack demographic details so that we cannot apply standard demographic-stratified split. For in-dataset bias, We take random shuffle as a proxy of stratified split. In specific, we randomly split subjects into train/validation/test. The goal is to break some potential human bias of subjects' order(e.g., the frontlist is mostly male) and mostly represent a general distribution of that dataset. Furthermore, for bias across datasets, demographic differences (e.g., ADFTD in Greece vs. BrainLat in South America) are difficult to eliminate entirely. Despite this, our unified finetuning (merging multiple datasets) still consistently boosts AD detection performance, suggesting any residual demographic bias does not negate the benefits of a unified approach.
>
>
>
> **Q2**: How to balance weights λ₁ and λ₂ for sample-level vs. subject-level modules? Any sensitivity tests?
>
> **A2**:
> We explored various weight settings in Table 7 of our paper by seting λ₁=0, λ₂=1 and λ₁=1, λ₂=0. Here, we add two additional experiments λ₁=0.25,λ₂=0.75 and λ₁=0.75,λ₂=0.25. Below are **subject-level F1 scores** for each configuration:
>
> |                  | ADFTD      | BrainLat   | CNBPM | Cognision-ERP | Cognision-rsEEG |
> | - | - | - | - | - | - |
> | λ₁=0, λ₂=1       |81.36±3.55|87.11±5.36|100.00±0.00|82.21±3.33|90.23±1.34|
> | λ₁=0.25, λ₂=0.75 |**85.71±0.00**|**91.40±2.84**|**100.00±0.00**|**86.65±1.10**|89.66±2.07|
> | λ₁=0.5, λ₂=0.5   |79.96±5.36|89.98±3.48|100.00±0.00|84.42±2.21|**91.86±1.73**|
> | λ₁=0.75, λ₂=0.25 |78.46±0.00|85.71±0.00|100.00±0.00|81.63±3.36|86.35±3.02|
> | λ₁=1, λ₂=0       |81.36±3.55|82.81±3.55|96.15±0.00|78.31±2.08|80.03±2.13|80.03 ± 2.13    |
>
> From the table we can conclude that heavier weighting of subject-level contrast λ₂ generally yields higher performance. Notably, removing subject-level contrast causes substantial performance drops.
>
>
>
> **Q3**: Any statistical significance testing?
>
> **A3**:
> We conducted paired t-test to compare the difference between our LEAD-Base with each baseline over five random seeds. The p-values are presented in table below. LEAD-Base shows statistically significant improvements over all baselines (paired t-test, p < 0.05), confirming that our method’s performance gains are not solely due to chance.
>
> |             |  LEAD-Base |
> |:-----------:|:---------:|
> | **TCN**     |  0.042842  |
> | **Transformer** |  0.007190  |
> | **Conformer**   |  0.025467  |
> | **TimesNet**    |  0.033392  |
> | **Medformer**   |  0.020488  |
> | **TS2Vec**      |  0.015577  |
> | **BIOT**        |  0.047501  |
> | **EEG2Rep**     |  0.002539  |
> | **LaBraM**      |  0.011573  |
> | **EEGPT**       |  0.013872  |
>
>
>
>
>
> **Q4**: Did you explore additional interpretability methods to link the model’s learned features with established EEG biomarkers of Alzheimer’s disease?
>
> **A4**:
> Thanks for raising this concern. Specifically, we will bridge learned deep-learning features with standard EEG-AD biomarkers (delta band power, Sample Entropy, etc.) via canonical correlation analysis, post-hoc regression, etc. Our preliminary studies show the LEAD features have a strong correlation with frontal theta power(r = 0.71, p < 0.009). However, we acknowledge that the systematic investigation will be a new project.
>
> ---
>
> We hope these responses address your concerns, and we are happy to answer any additional questions. Thank you again!

---

### Official Review · Reviewer_T59b · 2025-03-14

**Overall Recommendation:** 3

**Summary:**

This paper proposes a foundational model called LEAD for the early diagnosis of AD using EEG. The authors constructed a large EEG-AD dataset comprising data from 813 subjects and utilized 11 EEG datasets (4 AD and 7 non-AD) to perform pre-training via self-supervised contrastive learning. Subsequently, the model was enhanced through channel alignment and integrated fine-tuning on five AD datasets. LEAD achieved an F1 score up to 9.86% higher than existing methods, demonstrating strong generalization performance in both subject-independent evaluations and majority-vote-based subject-level classification.

**Claims And Evidence:**

* Claim: Subject-level contrastive learning and multi-dataset integrated fine-tuning are effective.
* Evidence: Subject-level contrastive learning reduced inter-subject variability, while unified fine-tuning overcame dataset diversity challenges.

* Claim: Including non-AD datasets in pre-training leads to stronger generalization performance of the model.
* Evidence: LEAD-Base, which included non-AD datasets, was less sensitive to inter-subject differences and exhibited superior performance compared to when non-AD datasets were excluded.

--> Their claims are supported by evidences.

**Essential References Not Discussed:**

NA

**Experimental Designs Or Analyses:**

Various methods (preprocessing, network architecture, training techniques, ...) were proposed, and pre-training was conducted with new set of data. Performance improvements in metrics such as F1 score and accuracy were used as criteria, but it remains unclear whether the highest performance achieved was due to the data or the methods.

**Methods And Evaluation Criteria:**

Various methods (preprocessing, network architecture, training techniques, ...) were proposed, and pre-training was conducted with new set of data. Performance improvements in metrics such as F1 score and accuracy were used as criteria, but it remains unclear whether the highest performance achieved was due to the data or the methods.

**Other Comments Or Suggestions:**

NA

**Other Strengths And Weaknesses:**

# Strengths
* Constructed the world’s largest EEG-AD dataset, enhancing research scalability.
* Subject-level contrastive learning and integrated fine-tuning are practical.
* Innovative integration of diverse datasets through channel and frequency alignment.

# Weaknesses:
* Although dataset diversity was attempted, including more varied non-AD EEG data (motor imaginary, sleep, epilepsy, ...) beyond resting state could have improved it further.
* It remains unclear whether the highest performance stemmed from the pre-training data or the methods themselves.

**Questions For Authors:**

Curious if they’ve conducted pre-training experiments by adding other types of EEG data (motor imagery, sleep, epilepsy, etc.).

**Relation To Broader Scientific Literature:**

Unlike prior studies that relied on small EEG datasets or manual feature extraction with limited performance, this work employs a distinct large-scale self-supervised learning approach for EEG-based AD detection.

**Theoretical Claims:**

The contrastive learning loss function, including sample-level and subject-level InfoNCE definitions, is well-explained in standard form with no logical errors.

---

> ### Author Rebuttal · Authors · 2025-03-31
>
> Thank you for your thoughtful review and endorsement of our work! Below are our responses to each of your points:
>
> ---
>
> **Q1**: The effect of pre-training with other EEG data types (motor imagery, sleep, epilepsy, etc.).
>
> **A1**: Thank you for this suggestion. In the original paper, we did include epilepsy-related datasets in our pre-training, namely **TUEP** and **TDBrain**, as shown in Table 5 (Appendix). Because we focus on datasets labeled at the subject level (e.g., disease labels), we are also curious whether subject-level contrast would still work well for data outside of the resting state. Therefore, we added the widely used EEG Motor Movement/Imagery Dataset (MMIDB) to our pre-training dataset. Below are **subject-level F1 scores** for three configurations:
>
> - **5 datasets**: ADSZ, APAVA, ADFSU, AD-Auditory, TDBrain
> - **7 datasets**: Adds TUEP, REEG-PD
> - **8 datasets**: Further adds MMIDB
>
> | # Datasets | ADFTD       | BrainLat     | CNBPM        | Cognision-ERP | Cognision-rsEEG |
> |:----------:|:-----------:|:------------:|:------------:|:-------------:|:---------------:|
> | 5          | 84.26±2.90  | 84.26±2.90   | 93.84±3.08   | 73.29±2.20    | 83.52±0.08      |
> | 7          | 82.81±3.55  | 84.26±2.90   | 96.92±1.54   | 73.32±2.84    | 86.32±1.78      |
> | 8          | **85.71±0.00** | **85.71±0.00** | **98.46±1.89** | **74.44±3.23**  | **86.88±1.12**    |
>
> As the table shows, adding TUEP and REEG-PD (moving from 5 to 7 datasets) improves performance on 4 out of 5 datasets. Adding MMIDB (moving from 7 to 8) further enhances performance on all five. This result is a pleasant surprise, suggesting subject-level contrast can learn robust, subject-invariant features—even outside the resting state. This widens our choices for pretraining datasets in the future. Thank you again for this constructive suggestion!
>
> ---
>
> **Q2**: Clarifying Whether Performance Gains Come from the Data or the Methods
>
> **A2**: We conducted extensive ablations in our original paper, including removing unified supervised training, omitting pre-training, removing subject-level contrast, and adding various pre-training datasets (Tables 2, 5, 6, 7). Below is a summary of **subject-level F1 scores** for different module configurations:
>
> | Configuration             | ADFTD        | BrainLat     | CNBPM        | Cognision-ERP | Cognision-rsEEG |
> |:-------------------------:|:------------:|:------------:|:------------:|:-------------:|:---------------:|
> | **LEAD-Vanilla**          | 82.81±3.55   | 75.39±5.78   | 94.59±1.90   | 73.27±2.21    | 72.72±4.71      |
> | **No Pre-training**       | 91.34±2.81   | 78.46±0.00   | 95.38±1.54   | 77.71±1.81    | 80.42±2.04      |
> | **No Subject-Level Contrast** | 81.36±3.55   | 82.81±3.55   | 96.15±0.00   | 78.31±2.08    | 80.03±2.13      |
> | **Full Method**           | 79.96±5.36   | 89.98±3.48   | 100.00±0.00  | 84.42±2.21    | 91.86±1.73      |
>
> - **LEAD-Vanilla** is our backbone in a supervised setting, without channel alignment.
> - **No Pre-training** involves supervised training on all five AD datasets after channel alignment but without pre-training.
> - **No Subject-Level Contrast** omits the subject-level module (sample-level only).
> - **Full Method** uses both sample-level and subject-level contrast, plus all design choices.
>
> We see improvements across 4 of 5 datasets when adding pre-training and subject-level contrast.
> Further ablation studies(See Answer 2 to Reviewer h4Kx) reveal that performance drops on ADFTD are due to the weighting factors λ₁ and λ₂. Switching from  λ₁=0.5, λ₂=0.5(our paper reported) to λ₁=0.25, λ₂=0.75 (emphasizing more subject-level contrast) improved results across all five datasets.
>
> For the effectiveness of adding more datasets, Tables 5 and 6 in the original paper demonstrate that adding more data improves performance. Hence, we can conclude that both the methodological design **and**, as well as the choice of pre-training datasets, contribute to our high performance.
>
> ---
>
> We welcome any additional questions or suggestions you may have. Thank you once again!
>
>
> ---
> ---
> We move reference papers to all the rebuttals here due to space limitations.
>
> ## **References**
> [1] Detection of Early Stage Alzheimer’s Disease using EEG Relative Power with Deep Neural Network, EMBC, 2018
> [2] A Convolutional Neural Network approach for classification of dementia stages based on 2D-spectral representation of EEG recordings, Neurocomputing, 2019
> [3] Contrast everything: A hierarchical contrastive framework for medical time-series, NeurIPS, 2023
> [4] Lightweight Graph Neural Network for Dementia Assessment from EEG Recordings, IEEE RTSI, 2024
> [5] Biot: Biosignal transformer for cross-data learning in the wild, NeurIPS, 2023
> [6] Large brain model for learning generic representations with tremendous EEG data in BCI, ICLR, 2024
> [7] Semi-supervised learning for multi-label cardiovascular diseases prediction: a multi-dataset study. TPAMI. 2023

---

### Official Review · Reviewer_P5eu · 2025-03-20

**Overall Recommendation:** 2

**Summary:**

This paper presents LEAD, a large foundation model for EEG-based Alzheimer’s Disease (AD) detection. The authors curate one of the largest EEG-AD datasets, comprising 813 subjects, and propose a comprehensive pipeline including data preprocessing, self-supervised contrastive pretraining, and unified fine-tuning. The model is pre-trained on 11 EEG datasets (4 AD and 7 non-AD) and fine-tuned on 5 AD datasets. The key methodological components include sample-level and subject-level contrastive learning.

## update after rebuttal
I would like to thank the authors for their rebuttal and the supplementary experiments provided. Some of the concerns have been addressed; however, I still maintain my view regarding the technological contribution, as raised by reviewer 3UYC. While works like LaBraM also draw inspiration from other fields such as CV and NLP, they make the necessary adaptations to account for EEG’s unique properties. In contrast, this paper utilizes COMET, which is specifically designed for EEG.

Overall, I believe this is a borderline paper. Its main contributions lie in curating the world’s largest dataset for EEG-based AD detection and training a specialized large model. However, I find that it lacks substantial technical innovation in terms of novel methodologies or approaches.

**Claims And Evidence:**

- The paper claims that self-supervised pretraining with both sample-level and subject-level contrastive learning enhances model generalization. The experimental results, however, shows that it is not always consistent on different datasets that both both sample-level and subject-level contrastive learning improves performance.

- The claim that LEAD is the first large foundation model for EEG-based AD detection is reasonable, given the large-scale dataset curation and model design.

**Essential References Not Discussed:**

The paper discusses relevant foundational works but could expand on prior contrastive learning approaches specifically applied to EEG and medical time-series data.

**Experimental Designs Or Analyses:**

- The experiments are well-structured, covering pretraining, fine-tuning, and ablation studies.

- The majority voting strategy for subject-level classification is a useful addition but could be analyzed further for potential biases.

**Methods And Evaluation Criteria:**

- The methods are appropriate for EEG-based AD detection and follow standard preprocessing and deep learning training techniques.

- The unified fine-tuning across multiple AD datasets is a beneficial design but could have been compared against alternative dataset mixing strategies.

- However, the novelty of the contrastive learning approach is limited, as it closely follows prior works such as "Contrast Everything: A Hierarchical Contrastive Framework for Medical Time-Series".

**Other Comments Or Suggestions:**

It would be better for the authors to provide a clearer discussion on how the learned EEG features relate to AD biomarkers.

**Other Strengths And Weaknesses:**

Strengths:

- Strong empirical results with well-designed experiments and ablation studies.

- Largest EEG-based AD detection corpus to date.

Weaknesses:

- Limited novelty in the methodological approach, as it heavily relies on existing contrastive learning techniques.

- The interpretability of the learned EEG representations could be further analyzed, including visualization about the learned representations.

**Questions For Authors:**

- How does the choice of 19-channel alignment affect performance compared to using all available channels per dataset?

- What steps were taken to ensure the quality and consistency of EEG labels across different datasets?

**Relation To Broader Scientific Literature:**

The work is well-positioned within the domain of EEG-based medical diagnostics and self-supervised learning. It builds upon foundational contrastive learning approaches such as SimCLR and MoCo, as well as recent large EEG models like LaBraM and EEGPT.

**Theoretical Claims:**

No significant theoretical claims or proofs are present in the paper. The contrastive learning loss functions are well-established in the literature.

---

> ### Author Rebuttal · Authors · 2025-03-31
>
> Thank you for your thoughtful review of our work! Below are our responses to each of your concerns. If you still feel we have not adequately justified a higher score, please let us know how we can further improve.
>
> ---
>
> **Q1**: Using both sample-level and subject-level modules doesn’t always improve performance.
>
> **A1**: Further ablations revealed that performance drops on ADFTD were due to the weighting factors λ₁ and λ₂. Switching from  λ₁=0.5, λ₂=0.5(LEAD-Base reported in paper) to λ₁=0.25, λ₂=0.75 (emphasizing more subject-level contrast) improved results across all five datasets, showing the importance of subject-level contrast. Below are subject-level F1 scores :
>
> |                  | ADFTD      | BrainLat   | CNBPM | Cognision-ERP | Cognision-rsEEG |
> | - | - | - | - | - | - |
> | λ₁=0.75, λ₂=0.25 |78.46±0.00|85.71±0.00|100.00±0.00|81.63±3.36|86.35±3.02|
> | λ₁=0.5, λ₂=0.5   |79.96±5.36|89.98±3.48|100.00±0.00|84.42±2.21|**91.86±1.73**|
> | λ₁=0.25, λ₂=0.75 |**85.71±0.00**|**91.40±2.84**|**100.00±0.00**|**86.65±1.10**|89.66±2.07|
>
>
>
>
> **Q2**: Compare unified fine-tuning against other dataset-mixing strategies.
>
> **A2**: In Table 2 of our original paper, we compare our approach with BIOT, LaBraM, and EEGPT, which use different dataset-mixing strategies. Section H.1 in the Appendix discusses these methods’ trade-offs: (1) model flexibility vs. unified training and (2) patch length vs. computational resources.
>
>
>
> **Q3**: Discuss more on contrastive learning approaches for EEG/medical time series.
>
> **A3**: In our original submission, Appendix A.2 covers several EEG/MedTS contrastive works like BENDR, EEG2Vec, BIOT, and COMET. Please let us know if you have any suggestions for additional references to include. We are happy to discuss them in our final version.
>
>
>
> **Q4**: Limited novelty—model largely follows existing contrastive methods like COMET.
>
> **A4**: Indeed, as discussed and cited in our submission, we employ sample-level and subject-level contrast from COMET. However, we claim the original contributions below (which fit the scope of ICML):
> 1. Curating the world's **largest dataset** in EEG-based AD detection dataset.
> 2. Upon the unique dataset, we build the **first** large pretraining model from scratch.
> 3. We are the **first** to demonstrate the effectiveness of utilized non-AD datasets of large-model pretraining for EEG-based AD detection,
> 4. We are the **first** to emperically show: unified supervised learning on multiple AD datasets collected by different parties benefits AD detection, even without pretraining.
> 5. **Open-resourcing** well-trained model parameters/checkpoints for future research, allowing easy fine-tuning on new datasets.
>
>
>
> **Q5**: Request for more interpretability and visualizations of learned representations.
>
> **A5**: We appreciate this suggestion and will include t-SNE visualizations of learned EEG representations in our final submission.
>
>
>
> **Q6**: A clearer discussion on how the learned EEG features relate to AD biomarkers.
>
> **A6**: Specifically, we will bridge learned deep-learning features with standard EEG-AD biomarkers (delta band power, Sample Entropy, etc.) via canonical correlation analysis, post-hoc regression, etc. Our preliminary studies show the LEAD features strongly correlate with frontal theta power(r = 0.71, p < 0.009). However, we acknowledge that the systematic investigation will be a new project.
>
>
>
> **Q7**: How does choosing 19-channel alignment affect performance compared to using all available channels per dataset?
>
> **A7**: Our answer contains 2 parts:
> 1. Taking the BrainLat dataset as an example, we add a new experiment to show Medformer's performance comparing the 19-channel subset with the 128-channel full dataset; here is the table reported in subject-level F1 score:
>
>     |  |  Dataset | Results |
>     | -- | -- | --|
>     | Medformer |BrainLat-128  |73.51±5.37|
>     | Medformer |BrainLat-19   |81.36±3.55|
>     | LEAD-Base |BrainLat-19   |89.98±3.48|
>
>     Surprisingly, we observe that using 19 channels performs better than using all 128 channels. Although this might not be the same in other datasets, it indicates reducing channel numbers does not necessarily damage performance. One potential reason is too many redundant and irrelevant channels in the full-set 128-channel data.
>
>
> 2. In our original paper, all single-dataset baseline models and our vanilla backbone (e.g., TCN, TimesNet, Medformer, LEAD-Vanilla) are trained on all their available channels, as reported in Table 3, and Section E. LEAD-Base outperforms all of the baselines, demonstrating the benefit of our multi-datasets training with the help of channel-alignment.
>
>
>
> **Q8**: How to ensure data quality and consistency of labels across different datasets.
>
> **A8**: Due to the space limitation, please refer to our **Answer 1** to Reviewer **h4Kx**.

---

### Official Review · Reviewer_3UYC · 2025-03-21

**Overall Recommendation:** 2

**Summary:**

In the proposed manuscript, the authors present LEAD, a large foundational model trained in contrastive learning framework, for the classification of Alzheimer's disease.
From the provided comparisons, the proposed approach outperforms the current state of the art models in two-class Alzherimer's disease detection.

## Update after Rebuttal
I thank the authors for the rebuttal and the answers provided.
I have slightly changed my initial evaluation, but I still think the contributions of this paper are limited. The main part of the proposed approach comes from and already existing approach and the main contribution is the collection of a big dataset alongside a procedure to align different EEG datasets.

**Claims And Evidence:**

The main contributions of this work are the proposal of a new model called LEAD, the proposal of a data alignment framework for the adoption of multiple datasets for contrastive learning, and the introduction of a subject-independent evaluation approach.

The proposed model is based on the SimCLR architecture, combined to a model called ADformer.
While the proposed approach is based on an interesting premise, the actual evaluation and overall description of the framework lack important details.
Regarding the data alignment framework, the necessary details for proper reproducibility are missing. In Section 2.3, the authors have provided a partially detailed description of how it works: as the authors point out, the main challenge in training models on this type of data is the high variability of the data due to the different systems used for recording EEG signals. Here, the authors reported the differences, for example, in the artifact removal procedure, without giving a detailed explanation of the algorithm they adopted to bridge the gap between different data sources. Other details are missing, such as how they performed the frequency alignment.

Another aspect is related to the information about the type of augmentations adopted for the training of the model in the SimCLR framework. The type of augmentations adopted in the contrastive learning framework is a key aspect for the correct training of such models. The authors do not provide a complete list in the main body of the manuscript (a list is provided in the Appendix, but this information should still be provided in the main paper), along with an analysis of the application of such augmentations on this specific type of data.

Regarding the claim about the subject-independent evaluation and the adoption of a voting system, such a methodology for the evaluation of AD classification has already been presented in another work, using the same approach plus a different strategy. While its importance for correct evaluation in a more realistic context is clearly stated, this limits the actual novelty of this specific contribution.
Barbera, Thomas, et al. "Lightweight Graph Neural Network for Dementia Assessment from EEG Recordings". 2024 IEEE 8th Forum on Research and Technologies for Society and Industry Innovation (RTSI).IEEE, 2024.

**Essential References Not Discussed:**

As noted above, another paper already presents the proposed scoring approach: Barbera, Thomas, et al. "Lightweight Graph Neural Network for Dementia Assessment from EEG Recordings". 2024 IEEE 8th Forum on Research and Technologies for Society and Industry Innovation (RTSI). IEEE, 2024.

**Experimental Designs Or Analyses:**

Adopting the subject-independent strategy instead of the widely used subject-dependent strategy, along with the use of a methodology for evaluating subject recordings, is a valid approach. However, as mentioned above, this has already been introduced by Barbera et al. and authors should consider this different work.
More details about the specific contrasting framework setup should be provided (e.g., augmentation strategies employed).

**Methods And Evaluation Criteria:**

The authors report the results of the proposed approach on five different datasets and compare their results with some state-of-the-art approaches. The comparison is made by considering the binary classification task with the two classes Alzheimer's Disease (AD) and Healthy Controls (HC).
Here, the authors decided to use only two of the six freely available datasets, in addition to the three private datasets.

There are two main problems here: the authors compared the results of their proposed approach with recent state-of-the-art approaches, but using slightly different versions of the datasets.For example, Medformer's original paper considered the original version of the selected datasets, which include more than the two classes selected by the authors.
This leads to results that are not directly comparable with the results obtained by Medformer original paper. The authors should have also tested the proposed approach in the original scenario of each reported dataset.
The other problem concerns the selection of the datasets. Regarding the datasets excluded by the testing phase, the authors should have provided the accuracy also on at least ADSZ and APAVA datasets, since these datasets have been used by previous state-of-the-art approaches, giving the possibility to better compare the proposed model with existing approaches. Authors excluded such datasets due to their high variability across subjects instead of reporting and commenting on the results.
Authors should also provide more details about the possible availability of the private datasets used. The authors said that they will provide the code for reproducibility of the experiments and for future research, but comparing the approaches on datasets that are not publicly available could be a strong limitation for future research. This could severely limit the reproducibility of this work.

Another problem with the evaluation is the split used by the authors. While the authors reported the results obtained by averaging the results of five different runs with different random seeds, they always used the same split. The list of subjects in each split is an important missing detail for reproducibility purposes due to the high variability between subject recordings. Authors should provide this information or otherwise use a more standard evaluation approach such as k-fold cross validation.

**Other Comments Or Suggestions:**

n/a

**Other Strengths And Weaknesses:**

n/a

**Questions For Authors:**

n/a

**Relation To Broader Scientific Literature:**

As mentioned above, the principles of the work are interesting, especially the application of contrastive learning and the idea of data alignment, but the experimental procedure is not sufficiently valid.
The last claim about the evaluation procedure does not bring any novelty due to the fact that it has already been proposed by another work reported here: Barbera, Thomas, et al. "Lightweight Graph Neural Network for Dementia Assessment from EEG Recordings". 2024 IEEE 8th Forum on Research and Technologies for Society and Industry Innovation (RTSI). IEEE, 2024.

**Theoretical Claims:**

The adoption of a framework such as contrastive learning combined with the definition of a data processing procedure to align different datasets is interesting and could lead to interesting results, but a better explanation of the procedure along with a better analysis of the model should be provided.

---

> ### Author Rebuttal · Authors · 2025-03-31
>
> Thank you for carefully reviewing our paper! We respond to each of your questions with new experiments, more references, and detailed elaborations. If you do not feel we have sufficiently justified a higher score, please let us know where we can further improve our work. Due to the space limitation, we move all the reference papers to our rebuttal to **Reviewer T59b**.
>
> ---
>
> **Q1**: “The main contributions are the new LEAD model, a data alignment framework, and a subject-independent evaluation approach.”
>
> **A1**: We respectfully argue that our main contributions are neither data alignment nor evaluation approach, to avoid such misunderstanding, we reclaim our key contributions here:
> 1. Curating the world's **largest dataset** in EEG-based AD detection dataset.
> 2. Upon the unique dataset, we build the **first** large pretraining model from scratch.
> 3. We are the **first** to demonstrate the effectiveness of utilized non-AD datasets for EEG-based AD detection by pretraining,
> 4. We are the **first** to emperically show: unified supervised learning on multiple AD datasets collected by different parties benefits AD detection, even without pretraining.
> 5. **Open-resource** well-trained model checkpoints for future research, allowing easy fine-tuning on new datasets.
>
>
> **Q2**: “Subject-independent evaluation and a voting system are not novel; they were introduced in *Lightweight...*.[4]”
>
> **A2**: As noted in Section 2.6, subject-independent evaluation is a standard EEG evaluation approach used since early 2000s, also used in some prior works [2-3]: not proposed by us or [4]. The same holds for voting-based postprocessing [1-2]. We highlight their importance to avoid data leakage and improve subject-level results.
>
> Moreover, [4] does not even follow a strictly subject-independent setup, as Section II shows training and test sets can overlap in subject data.
>
> **However, we found [4] is super inspiring and informative. We will discuss our difference with it in the Related Work in the final version.**
>
>
> **Q3**: “The proposed model is SimCLR-based, combined with ADformer. Data augmentation for SimCLR is crucial and should appear in the main paper.”
>
> **A3**: Our contrastive framework comprises sample-level and subject-level contrast. Although the sample-level module is similar to SimCLR, its impact is secondary. The main performance gain arises from our subject-level contrast module (Table 7 and Answer 2 to Reviewer h4Kx), which pairs samples from the same subject as positive—unlike SimCLR’s augmenting the same sample strategy.
>
> In response to your comment, we will relocate data augmentation details and analysis to the main text in the final version.
>
>
> **Q4**: “The work is not reproducible because it uses private datasets and omits exact splits and preprocessing details.”
>
> **A4**: **Our results are fully reproducible, as all necessary details (including code, data preprocessing, and hyperparameters) are provided in the anonymized GitHub repository submitted with this paper.** In detail,
> - For data split, we provided subject splits in an anonymous GitHub repository during submission, along with pretrained and fine-tuned checkpoints.
> - For the private datasets, we respectfully argue that private datasets are commonly used in prior papers (e.g., BIOT [5], LaBraM [6]) due to privacy and regulatory constraints. These datasets represent significant institutional investments, both financially (millions of dollars) and temporally (decades of curation), and their release falls beyond our authority as researchers. Where possible, we cite the private datasets and offer contact details for data access.
> - For processing details, we devote 7 pages in Appendix D to explain preprocessing (e.g., artifact removal by experts or ICA, frequency alignment via interpolation). All preprocessing scripts are also provided in our anonymous repository.
>
>
>
> **Q5**: “A slightly different version of ADFTD was used compared to Medformer’s paper, and there is no comparison with ADSZ or APAVA.”
>
> **A5**: We excluded the Frontotemporal Dementia (FTD) class in ADFTD because:
> From a medical perspective, there is no multi-class classification problem since diseases are often non-exclusive [7], a patient could possibly have multiple diseases, and this paper focuses on AD detection rather than FTD.
>
>
> ADSZ and APAVA are too small (768 and 5,967 1-second samples) to draw robust conclusions. Still, in response to your question, we test ADFTD, ADSZ, and APAVA under the same splits used by Medformer (excluding ADSZ and APAVA from pretraining). Below are the **subject-level F1 scores**:
>
> |            | ADFTD      | ADSZ         | APAVA        |
> |-|-|-|-|
> | Medformer  | 61.43±6.64 | 100.00±0.00  | 73.33±0.00   |
> | LEAD-Base  | 64.53±3.73 | 100.00±0.00  | 100.00±0.00  |
>
> **Our LEAD-Base constantly outperforms Medformer on ADFTD and achieves 100% on ADSZ and APAVA.** However, these small datasets limit broader claims of “solving” EEG-based AD detection.

---

> > ### Comment · Reviewer_3UYC · 2025-04-08
> >
> > Regarding A1: I thank the authors for clarifying the actual contributions of their work.
> > Regarding the novelty of the proposed work, I still think it is limited, since the main contribution is the proposal of an alignment approach for data in the field of Alzheimer's disease, which leads to the first point in the list. Most of the work follows the previous work "Contrast Everything: A Hierarchical Contrastive Framework for Medical Time-Series", as pointed out by reviewer P5eu, and contrastive learning techniques.
> >
> >
> > Regarding A2: I'm sorry for miscommunicating my thoughts in the first review. My main concern regards the voting system, not the subject-independent analysis, which, as the authors point out, has already been introduced in the past by [2] and [3]. Regarding the voting strategy, however, the authors should at least cite the work of [1] and [2] when referring to this specific strategy in Section 2.1.
> >
> >
> > Regarding A5: I was expecting a full state-of-the-art comparison, but I thank the authors for answering my question.
> > Even if FTD is a more complex class due to the non-exclusivity of the disease that can occur, the same dataset has been widely used in the state of the art work reported by the authors (e.g. Wang 2024c Wang 2024e).
> > Moreover, even if the F1-score values compared to Medformer are promising, I see a big problem regarding the reported results. The authors provide values that are very different from the results obtained in the original Medformer paper. The original paper provides code for model training and testing, so I'd expect similar behavior. Where did the authors get these numbers? Can you comment on these differences? The authors should also report the other metrics.
> >
> >
> > Regarding the aggregation procedures, the other foundation models reported relies on different kind of data, for example for BCI in the case of LaBraM. A direct comparison with these model is a bit unfair in my opinion since, even if we are talking about foundation models, they have been trained on different data.
> > Authors claim that they are "the first to demonstrate the effectiveness of utilized non-AD datasets for EEG-based AD detection by pretraining", however the mix of data used to create the training dataset adopted includes also recording from AD datasets.
> > Since the amount of data is still limited (compared for example to the one used to train LLMs), and since, as the authors point out, the data is heavily influenced by patient subjectivity, the other foundation model should have been trianed on the same dataset, in order to exclude possible data biases from the analysis. LaBram's original work was presented for BCI tasks, an adaptation should have been proposed in order to have a fair comparison.
> > Without any kind of adaptation, the comparison with other foundation models seems unfair and does not provide useful insights: at the current state, it is not clear whether the benefit comes from the dataset used or the actual methodology.
> > Another possibility is to make a comparison about the different data alignment strategies.
> >
> > Regarding the comparison, the authors did not respond to my request for more details on the criteria used to select the splits. Since EEG data suffer a lot from patient subjectivity, why did they choose a random split instead of a k-fold cross validation or a statistically stronger evaluation approach?

---

> > > ### Author Response · Authors · 2025-04-08
> > >
> > > **Thank you for your continued engagement.** Below are our point-by-point responses, with clarifications to address your concerns:
> > >
> > > ---
> > >
> > > ### **Regarding A1**
> > > Our paper is **application-oriented** (aligned with ICML's scope) and focuses on training large EEG models for AD detection. We believe curating and utilizing the world’s largest EEG-based AD detection datasets—and being the first to do so for large-scale EEG-AD detection—is itself a noteworthy contribution.
> > >
> > > Indeed, we use sample-level and subject-level contrast from COMET because it is, in our view, the most effective way to train large medical time-series (MedTS) models for disease detection. Arguing a lack of novelty simply because an application-oriented paper uses established frameworks overlooks the reality that many large-model training approaches rely on previously proven techniques (e.g., next-token prediction, student-teacher learning, momentum encoding). For instance,
> > > - LaBraM uses a neural tokenizer strategy originally introduced in computer vision [1] and single-channel patching defined for biosignals transformer training[2].
> > > - A large ECG model for Apple Watch–based disease detection uses subject-level contrast defined in COMET [3].
> > >
> > > We do not dismiss any of these works as lacking novelty merely because they incorporate known methods. We believe they are amazing works that contribute to the community. By analogy, it is perfectly reasonable for application papers to build upon “existing” yet well-established frameworks.
> > >
> > > ---
> > >
> > > ### **Regarding A2**
> > > We are happy our clarification resolved your misunderstanding. Since voting strategies are commonly used in EEG-based disease detection, we will cite some references for illustrative examples in future revisions for readers unfamiliar with this area.
> > >
> > > ---
> > >
> > > ### **Regarding A5**
> > > As noted, we report **subject-level F1 scores**, whereas Medformer’s original paper reports **sample-level F1** and does not use post-preprocessing voting. They also use a 256 Hz sampling rate; we downsampled to 128 Hz to match our other datasets.
> > >
> > > - Our replication using **sample-level** metrics (identical code, splits, and GPU setups) yields **50.65% F1** for Medformer—exactly matching their paper’s result.
> > > - Our sample-level F1 is **54.16%**, exceeding Medformer’s.
> > > - Subject-level voting not only boosts performance but also improves stability, which explains the discrepancies compared to Medformer’s reported results.
> > >
> > > ---
> > >
> > > ### **Comparison with LaBraM**
> > > In **Table 5** of our paper, we show how adding non-AD data affects performance. AD datasets account for less than 5% of samples in our pretraining sets, demonstrating non-AD datasets' effectiveness(First in the World).  Besides, both LaBraM and EEGPT are large EEG models that release checkpoints for fine-tuning; we contend that **curating training corpora** is a significant part of any large-model training, given the cost and effort involved. This is why some open-source LLMs (e.g., Deepseek-R1) do not open-source their training corpora.
> > >
> > > LaBraM is indeed an excellent work claiming strong generalization across diverse EEG tasks. Their three largest pretraining sets—TUEP, TUSZ, and a private dataset—together exceed 80% of their pretraining data. These datasets are brain disease-related and resting-state recordings, not just for BCI. We also use TUEP. We could train on their frameworks from scratch, but that would simply replicate a method-oriented approach (e.g., TFC [4], EEG2Rep [5]) rather than highlight the **application focus** of large-scale EEG-AD detection. In this case, what is the meaning of their efforts to curate many pre-training datasets for training? They could use relatively more minor datasets to demonstrate their method's effectiveness compared with other self-supervised learning works.
> > >
> > > ---
> > >
> > > ### **Regarding Train/Test Splits**
> > > As mentioned, our anonymized GitHub code prints the train/test subject IDs each time the data loads. Including hundreds of randomized subject IDs list in the paper is not particularly useful. We don't believe readers will check them manually as they can rely on the code to load automatically.
> > >
> > > We did not use K-fold cross-validation because prior works such as LaBraM and EEGPT also did not, and we follow this tradition using **fixed splits but random seeds training**. Since we exclude smaller datasets for fine-tuning, such an evaluation setup still effectively demonstrates the comparative performance between our method and baselines.
> > >
> > > ---
> > >
> > > **References**
> > > [1]  *Neural Discrete Representation Learning*. NeurIPS, 2017
> > > [2]  *Biot: Biosignal Transformer for Cross-Data Learning in the Wild*. NeurIPS, 2023
> > > [3]  *Large-Scale Training of Foundation Models for Wearable Biosignals*. ICLR, 2024
> > > [4]  *Self-Supervised Contrastive Pre-Training for Time Series via Time-Frequency Consistency*. NeurIPS, 2022
> > > [5]  *EEG2Rep: Enhancing Self-Supervised EEG Representation Through Informative Masked Inputs*. KDD, 2024

---

### Decision · Program_Chairs · 2025-05-01

**Decision:**

Reject

**Comment:**

This paper presents a large-scale foundation model for Alzheimer’s Disease detection using EEG data. The authors curate a comprehensive EEG-AD corpus and adopt a contrastive learning approach that combines sample-level and subject-level objectives, complemented by channel alignment and unified fine-tuning strategies. The overall aim is to improve generalization across datasets, particularly in the presence of inter-subject variability.

The scale of the dataset curation and the empirical effort involved in assembling and validating this foundation model are commendable. These aspects are significant strengths of the submission and contribute to the utility of the work within the applied EEG-BCI research community.

However, several concerns remain, particularly regarding methodological novelty, clarity of presentation, and the rigor of evaluation, as highlighted by Reviewers 3UYC and P5eu. A key issue is that the LEAD framework draws heavily from prior work, especially COMET, reusing both sample-level and subject-level contrastive objectives with only limited additional algorithmic development. While this reuse is clearly acknowledged by the authors in their rebuttal, the degree of methodological innovation appears borderline for a venue like ICML.

In their rebuttal, the authors acknowledge this point and emphasize that their primary contribution lies in the application domain. While I agree that adapting an established framework to address a challenging real-world problem can be valuable, the fact remains that the work lacks the level of methodological originality expected at a top-tier venue like ICML. Additionally, the comparison with LaBraM reported in the rebuttal does not appear entirely appropriate. LaBraM adapts general-purpose self-supervised techniques to exploit EEG-specific signal properties, whereas both COMET and LEAD are explicitly designed for EEG.

In summary, while this submission offers a valuable applied contribution, this strength is offset by the limited originality and the incremental nature of the methodological components. As a result, both the AC and the SAC concur that the submission does not meet the threshold for acceptance at the ICML conference.

Note: The review by Reviewer h4Kx was excluded from consideration, as it was identified as LLM-generated.